# Dietary flavanols preserve upper- and lower-limb endothelial function during sitting in high- and low-fit young healthy males

Alessio Daniele[1], Samuel J. E. Lucas[1,2] and Catarina Rendeiro[1,2]

[1]*School of Sport, Exercise and Rehabilitation Sciences, University of Birmingham, Birmingham, UK*
[2]*Centre for Human Brain Health, University of Birmingham, Birmingham, UK*

Handling Editors: Harold Schultz & Sophie Møller

The peer review history is available in the Supporting Information section of this article (https://doi.org/10.1113/JP289038#support-information-section).

**Abstract figure legend** Schematic indicating the design of an acute randomised, counterbalanced, double-blinded, cross-over, placebo-controlled human study in which 20 high-fit males and 20 low-fit males consumed either a low-flavanol (5.6 mg) or a high-flavanol (695 mg) cocoa drink prior to a 2 h uninterrupted sitting bout. This work established that intake of dietary flavanols prior to a sitting period preserves upper- and lower-limb endothelial function in both high- and low-fit males, as measured by flow-mediated dilatation (FMD) of the brachial and superficial femoral conduit arteries.

**Abstract** Sedentary behaviour is very prevalent in modern societies, with young adults spending approximately 6 h/day sitting. Sitting induces declines in endothelial function, increasing the risk of cardiovascular diseases. Nutrition during sitting may be used as a strategy to minimise such effects. This acute randomised, counterbalanced, double-blinded, cross-over, placebo-controlled human study investigated whether intake of dietary flavanols prior to a 2 h sitting bout could preserve upper- and lower-limb endothelial function in high- and low-fit individuals. Forty young healthy males (20 high fit; 20 low-fit) completed a 2 h sitting trial after consuming either a high-flavanol (150 mg (−)-epicatechin) or low-flavanol (<6 mg (−)-epicatechin) cocoa intervention. The primary outcome, superficial femoral artery flow-mediated dilatation (SFA FMD) and secondary outcomes, brachial artery (BA) FMD, resting shear rate, blood flow and blood pressure (BP) were assessed before and after sitting. Medial gastrocnemius microvasculature haemodynamics were assessed before, during and after sitting. Sitting significantly reduced FMD in the SFA and BA and increased diastolic BP in both fitness groups. The high-flavanol intervention prevented FMD declines in both arteries, with no effects on BP. Sitting significantly decreased shear rate and blood flow in both arteries in both fitness groups, with no effects of the flavanol intervention. Sitting further resulted in declines in tissue oxygenation (TOI), detectable within 10 min, and impaired TOI desaturation and speed of reperfusion during hyperaemia 2 h post-sitting, with no effects of flavanols. Flavanol-rich foods may be efficacious at counteracting sitting-induced endothelial dysfunction during prolonged sitting in both high- and low-fit individuals.

(Received 10 April 2025; accepted after revision 23 September 2025; first published online 12 October 2025)

**Corresponding author** C. Rendeiro: School of Sport, Exercise and Rehabilitation Sciences, University of Birmingham, Birmingham, B15 2TT, UK.    Email: c.rendeiro@bham.ac.uk

**Key points**

- Prolonged sitting temporarily impairs vascular function and blood pressure. Dietary strategies during sitting may ameliorate or aggravate the effects of sitting on vascular health.
- In this study, dietary cocoa flavanols consumed just before 2 h of uninterrupted sitting were effective at preventing sitting-induced reductions in vascular function in both the brachial and femoral arteries (as measured by flow-mediated dilatation) in young healthy men, with no effects on microvascular function or blood pressure.
- Flavanols were equally effective at preserving vascular function in individuals with high and low cardiorespiratory fitness during sitting.
- Individuals' high cardiorespiratory fitness did not protect against declines in vascular function and blood pressure during sitting.
- Consuming high-flavanol foods during sedentary periods may be used alone or in combination with other strategies (e.g. breaking up sitting) to reduce the impact of inactivity on the vascular system.

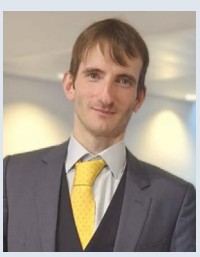

**Alessio Daniele** is a researcher with an overarching interest in vascular physiology. He has recently completed a PhD at the University of Birmingham (UK), investigating how lifestyle choices, such as fitness or diet, may modulate vascular health in the context of sedentarism. During this time, he has gained proficiency in ultrasound scanning of conduit arteries to assess human endothelial function. Prior to his PhD, he completed a bachelor's degree in Sport Science and an MSc in Health and Physical Activity at the University of Rome 'Foro Italico'. He further specialised, earning an MSc in Sport and Exercise Physiology from Bangor University. [Correction made on 31 October 2025 after first online publication: The author biography has been updated.]

## Introduction

Sedentary behaviour is extremely prevalent in modern societies. Over the past two decades, it is estimated that the prevalence of sedentary behaviours – particularly sitting time – increased from 5.5 to 6.5 h per day (equivalent to 18.2%) among young adults in the US (Yang et al., 2019). Sitting is a very common type of sedentary behaviour that includes leisure activities such as watching TV and playing video games, as well as occupational sitting and commuting (Park et al., 2020). A recent study has shown that more than half of young adults (10,808 adults, average age of 39.4 years old) spend at least 6 h per day sitting (Liao et al., 2024). For example, in the UK, the average adult spends the equivalent of 76 days per year in sitting activities (BHF, 2017). Sedentary time is positively associated with all-cause mortality (Ekelund et al., 2016; Gao et al., 2024) and it increases the risk of several chronic health conditions, including cardiovascular diseases (CVDs) (Li et al., 2022), cardiometabolic diseases (Koyama et al., 2020) and hypertension (Lim et al., 2017). A recent systematic review showed that sedentary behaviour is probably associated with excess healthcare costs in various parts of the world (Nguyen et al., 2022). Specifically in the UK, the direct healthcare costs of sitting for more than 6 h per day were estimated over a 1 year period to be approximately £0.7 billion (Heron et al., 2019).

Experimental human studies have shown that uninterrupted prolonged sitting has detrimental effects on endothelial function in young healthy adults (Morishima et al., 2017; O'Brien et al., 2019; Thosar, Bielko, Mather, et al., 2015). Endothelial function can be accurately measured by flow-mediated dilatation (FMD), which is a non-invasive ultrasound-based technique for assessing nitric oxide (NO)-dependent vascular function in peripheral conduit arteries in humans (Thijssen et al., 2019). Concerningly, even a few hours of uninterrupted prolonged sitting (1–6 h) can reduce FMD in young healthy adults, particularly in lower-limb arteries (reviewed in Daniele et al., 2022). As shown in a recent systematic review and meta-analysis, depending on the artery investigated, sitting-induced declines in lower-limb FMD can reach 1.75–2.51% (Paterson et al., 2020). It is well established that athletes have a better cardiovascular system and a lower risk of developing cardiovascular-related pathologies than the general population (Runacres et al., 2021; Štursová et al., 2023). Plausibly, individuals with a higher level of cardiorespiratory fitness may experience attenuated sitting-induced reductions in vascular function. However, it is unclear whether cardiorespiratory fitness plays a role in modulating vascular responses following acute exposure to prolonged sitting. The literature is limited, and results are conflicting, with one study showing that sitting-induced endothelial dysfunction in the popliteal artery is prevented in aerobically fit individuals (Morishima et al., 2020), while another more recent study has shown no protective effects during sitting from being aerobically fit (Liu et al., 2021).

Several activity-based interventions are effective at preventing/attenuating the acute sitting-induced reductions in lower-limb FMD, including walking breaks (Thosar, Bielko, Mather, et al., 2015), stair-climbing breaks (Cho et al., 2020), running/cycling preceding sitting (Ballard et al., 2017; Morishima et al., 2017) and intermittent leg fidgeting (Morishima et al., 2016). Nutritional strategies, however, have not been fully explored in the context of sitting, with only a few acute studies suggesting that vitamin C and beetroot juice may be beneficial at protecting lower-limb vascular function during 3 h of uninterrupted sitting (Morishima et al., 2022; Thosar, Bielko, Wiggins, et al., 2015). Interestingly, eating and sitting are often concurrent behaviours. As such, healthy food choices during periods of prolonged sitting might be an effective and ecologically valid strategy to attenuate or rescue the negative impact of sitting on the peripheral vascular system.

In the last few decades, both observational and epidemiological studies have suggested that diets rich in polyphenols, and particularly flavonoids, a subgroup of plant-derived compounds present in fruit and vegetables, reduce the risk of future CVD (Micek et al., 2021; Sesso et al., 2022). This has been largely confirmed by human randomised controlled trials demonstrating that flavonoid-rich foods (e.g. berries, cocoa, citrus fruits) improve endothelial function both acutely (1–7 h), as well as short- to long-term (from 2 weeks to 6 months) across the lifespan (Deng et al., 2024; Dicks et al., 2022; Jalili et al., 2024; Raman et al., 2019; Sun et al., 2019). For example, regular consumption of cocoa flavanols results in improvements in brachial FMD of more than 1% (Sun et al., 2019), which can translate into a reduction of risk of future CVD events by at least 9% (Green et al., 2011). Importantly, cocoa flavanols can induce rapid, transient improvements in brachial endothelial function within 1 h of ingestion (Sansone et al., 2017; Schroeter et al., 2006), as circulating flavanol metabolites, particularly (−)-epicatechin-derived (Borges et al., 2018; Ottaviani et al., 2016), stimulate the arterial endothelium to increase NO bioavailability and reduce endothelin-1 production (Loke et al., 2008; Moreno-Ulloa et al., 2014, 2015). Importantly, a recent study further suggests that the acute beneficial effects of cocoa flavanols may extend to the lower-limb arteries, making it particularly relevant in the context of sitting-induced lower-limb endothelial dysfunction (Bapir et al., 2022). Interestingly, there is a considerable variation in how individuals benefit from flavonoid intake (Cassidy & Minihane, 2017; Manach et al., 2017) and currently it is unclear which individual

characteristics may maximise that benefit. For example, recent data indicate that individuals with a low-quality diet, including low intake of flavonoids, may benefit more from flavonoid consumption (Brickman et al., 2023). To date, no studies have investigated the impact of cardio-respiratory fitness on vascular responses to flavonoids.

In the present study, we investigated whether acute intake of cocoa flavanols prior to a 2 h bout of uninterrupted sitting can be beneficial in improving endothelial function (as measured by FMD) in the lower-limb superficial femoral artery (SFA) and upper-limb brachial artery (BA) in high- and low-fit young healthy men. We further evaluated whether cocoa flavanols can preserve downstream lower-limb microvascular function and BP during sitting.

## Materials and methods

### Ethical approval

The study was conducted in accordance with the *Declaration of Helsinki*, and was approved by the University of Birmingham Science, Technology, Engineering and Mathematics ethics committee (ERN_19-0851). Informed written consent was obtained from all participants before enrolment in the study.

### Participants

Forty young healthy male adults (aged 18–45 years old), either 'high fit' (peak oxygen consumption [$\dot{V}_{O_2peak}$] $\geq$49 ml·kg$^{-1}$·min$^{-1}$, $N = 20$) or 'low fit' ($\dot{V}_{O_2peak}$ $\leq$41 ml·kg$^{-1}$·min$^{-1}$, $N = 20$) were recruited from the University of Birmingham (Birmingham, England) and the surrounding community. Prior to participation in the study, all participants provided a signed informed consent form and completed a general health and lifestyle questionnaire. Individuals were excluded from the study if they had a $\dot{V}_{O_2peak}$ between 41 and 49 ml·kg$^{-1}$·min$^{-1}$, or a history or symptoms of cardiovascular, renal, pulmonary, metabolic or neurologic disease, hypertension (BP higher than 140/90 mm Hg, according to recent guidelines (Mancia et al., 2023)), diabetes mellitus, anaemia, asthma, immune conditions or high cholesterol. In addition, smokers, individuals who were on weight-reducing diets or using prescribed/over-the-counter medications, or had recently undergone prolonged bed-rest periods, were also excluded from the study. Females were not included, as a separate female-only randomised controlled trial is required to ensure that interactions with distinct stages of menstrual cycle are addressed, given the rationale for differential modulation of vascular function by flavanols depending on the levels of circulatory oestrogen (Moreno-Ulloa et al., 2015, 2018).

### Study design

The study was a randomised, counterbalanced, double-blinded, placebo-controlled, cross-over intervention study (Fig. 1).

The study involved participation in a maximal oxygen consumption ($\dot{V}_{O_2max}$) test session to establish fitness eligibility and two experimental visits separated by at least 7 days. Following the assessment of cardio-respiratory fitness (based on $\dot{V}_{O_2peak}$), by performing a maximal aerobic capacity test on a cycle ergometer, participants were allocated to one of the two fitness categories: 'high fit' ($\dot{V}_{O_2peak}$ $\geq$49 ml·kg$^{-1}$·min$^{-1}$), and 'low fit' ($\dot{V}_{O_2peak}$ $\leq$41 ml·kg$^{-1}$·min$^{-1}$). Participants with a $\dot{V}_{O_2peak}$ between 41 and 49 ml·kg$^{-1}$·min$^{-1}$ were excluded from the study. These thresholds were established based on the American College of Sports Medicine (ACSM) $\dot{V}_{O_2max}$ norms for young men (ACSM, 2013), as representative of a 'poor/very poor' fitness level, and a 'good/excellent/superior' fitness level. If participants fitted the fitness inclusion criteria, a familiarisation ultrasound scan of the SFA and BA were performed, and they were invited to attend the laboratory for the two experimental visits.

Prior to the two experimental visits, participants were asked to fast for at least 12 h; they were also asked to refrain from caffeine, alcohol and polyphenol-containing foods/beverages (a detailed list of permitted foods/beverages and those to avoid was provided in advance), and any form of physical activity (above light intensity) for at least 24 h. Polyphenol metabolites may be detected in circulation for up to 80 h post-intake (González-Sarrías et al., 2017), but most metabolites are excreted within the first 24 h (Borges et al., 2018); as such, a polyphenol-restricted diet was maintained for 24 h prior to experimental visits to minimise the burden on participants. The two experimental visits started in the early morning (between 08:00 and 09:30), and volunteers were invited to rest in a supine position for approximately 15 min. To ensure and assess compliance with the 24 h dietary restrictions, all volunteers were contacted and reminded of what to eat/avoid 48 h prior to each visit and a 24 h dietary recall questionnaire was performed verbally at the beginning of each experimental visit. Pre-sitting/flavanol intervention baseline measures were assessed in this order: (1) resting upper arm systolic and diastolic BP, (2) resting heart rate, (3) SFA FMD (primary outcome measure), (4) medial gastrocnemius tissue oxygenation levels during reactive hyperaemia induced by SFA FMD (as assessed by near-infrared spectroscopy; NIRS) and (5) BA FMD. Immediately after baseline measures, the participant was asked to sit uninterrupted for 2 h on a comfortable chair and to consume (within 10 min) the high- or low-flavanol cocoa beverage (randomly assigned

and double-blinded). During the sitting trial, tissue oxygenation levels of the medial gastrocnemius (assessed by NIRS) were also recorded (Fig. 2). Flavanols were administered immediately prior to sitting to ensure their bioavailability during the 2 h sitting period, as flavanols are known to be bioavailable within 1 h of intake, peaking at 2 h post-ingestion (Schroeter et al., 2006).

While sitting, the participants were asked to avoid any lower body movements and to keep their legs parallel and bent at approximately 90° with both feet in a neutral position with the plantar surface on the floor, while having the trunk resting on the backrest. Volunteers were allowed to move their arms/hands (although instructed to avoid abrupt movements) to do activities such as computer typing, phone texting and writing. The sitting trial took place under the supervision of the investigator. Following two hours of sitting, the same outcome measures as described above were assessed in the same order (Fig. 2).

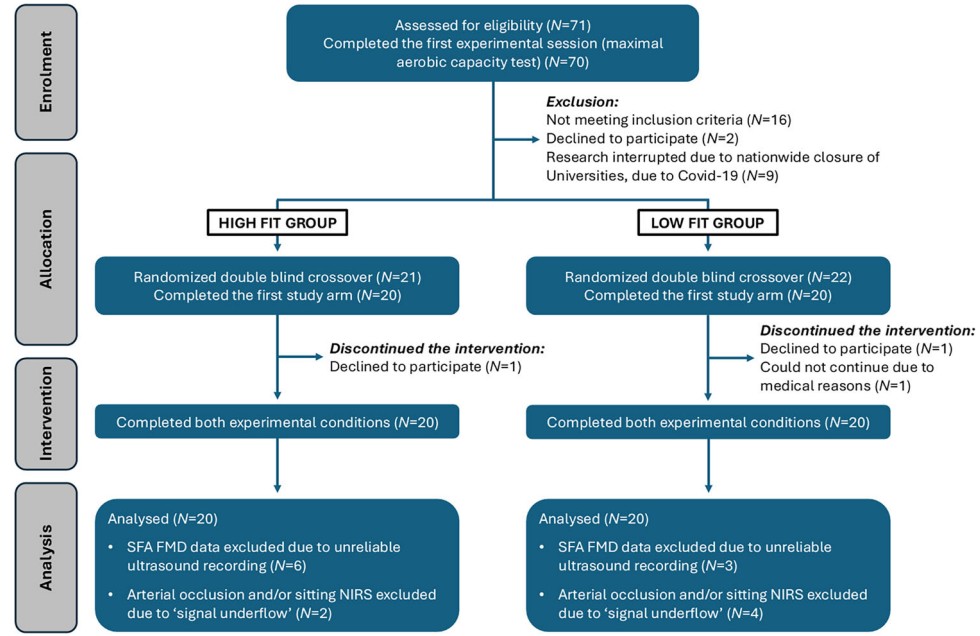

**Figure 1. Consolidated standards of reporting trials (CONSORT) flow diagram for the intervention study**
FMD, flow-mediated dilatation; NIRS, near-infrared spectroscopy; SFA, superficial femoral artery.

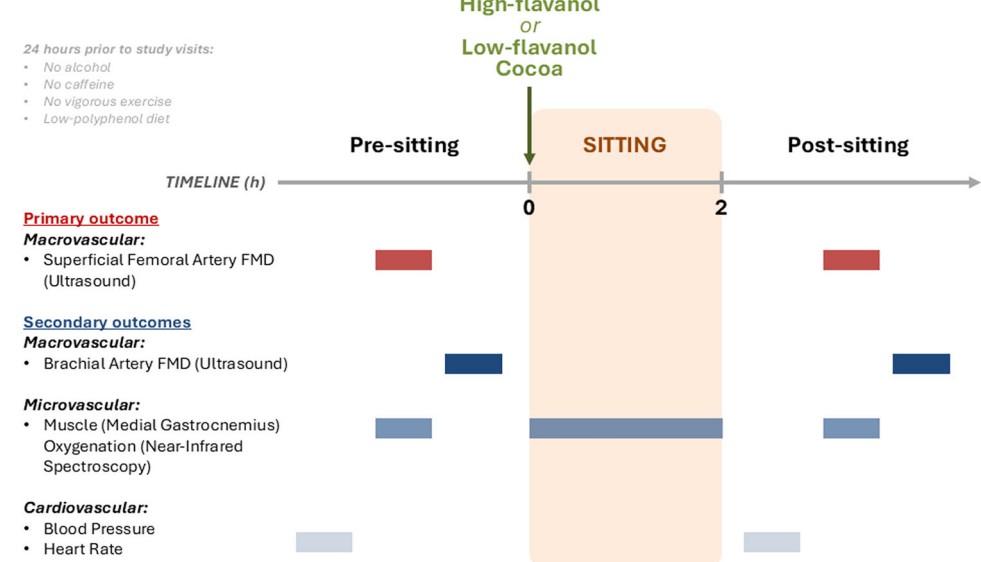

**Figure 2. Experimental study design featuring primary and secondary outcome measures assessed before and after a 2 h sitting trial**
FMD, flow-mediated dilatation.

Experimental visits were performed in a quiet, darkened and temperature-controlled laboratory (22–24°C), as recommended in FMD measurement guidelines (Harris et al., 2010; Thijssen et al., 2019).

### Habitual diet and physical activity

Participants also completed a food frequency questionnaire (FFQ) to assess habitual diet in the previous year and to wear a wrist-worn tri-axial accelerometer (GENEActiv, version 1.1, ActivInsights Ltd., Kimbolton, England) for seven consecutive days to measure habitual physical activity and sedentary/sitting time. Habitual dietary intake was assessed using the validated European Prospective Investigation into Diet and Cancer (EPIC) Norfolk FFQ (Bingham et al., 2001). The FFQ comprises 131 food products to select the intake frequency on a 9-point scale (never or less than once per month, 1–3 per month, once a week, 2–4 times per week, 5–6 times per week, once a day, 2–3 times per day, 4–5 times per day, and 6+ times per day) to estimate the habitual dietary intake within the past year. The FFQ EPIC Tool for Analysis (FETA) was used to calculate nutrient and food group data (Mulligan et al., 2014). To give a general overview of the habitual dietary intake, the following were reported: energy (kcal), fat (g), saturated fat (g), carbohydrate (g), sugars (g), fibre (g), protein (g), total flavonoids (mg) and portions of fruit and vegetables, calculated as one portion corresponding to 80 g, from National Health Service (NHS) guidelines (NHS, 2022a). Physical activity monitoring was measured using a tri-axial accelerometer, set to sample data at a measurement frequency of 85.7 Hz. After the data were collected, they were converted into a 60 s epoch and analysed using the GENEActiv software (GENEActiv, version 3.3, ActivInsights Ltd., Kimbolton, England).

### Aerobic capacity test

Participants' aerobic capacity was assessed by a maximal incremental exercise test on a cycle ergometer (Lode Excalibur Sport, Groningen, the Netherlands). Before commencing the exercise testing, preliminary measurements of participants' anthropometric characteristics were conducted using a telescopic measuring rod (Seca 220, Seca, Hamburg, Germany) and weighing scales (Champ II, Ohaus, Nanikon, Switzerland) for height and weight, respectively. The test began at 100 W and increased in increments of 30 W every 2 min until at least one of these conditions were met: (1) voluntary exhaustion, (2) rate of perceived exertion (RPE) score of 20 (Borg RPE scale: 6–20), or (3) not maintaining a cadence of ≥60 rpm. During the exercise testing, heart rate was monitored continuously using a wrist-worn

**Table 1. Nutritional composition of the high- and low-flavanol cocoa powders (12 g per individual dose)**

|  | Low flavanol | High flavanol |
| --- | --- | --- |
| Total polyphenols* | 260.0 | 1246.8 |
| Total flavanols (mg) | 5.6 | 695.0 |
| Procyanidins (dimers-decamers; mg) | ND | 459.6 |
| (−)-Epicatechin (mg) | <6 | 150.0 |
| (−) and (+)-Catechin (mg) | <6 | 85.4 |
| Theobromine (mg) | 278.4 | 262.8 |
| Caffeine (mg) | 22.2 | 27.6 |
| Fat (g) | 1.3 | 1.7 |
| Carbohydrate (g) | 1.2 | 2.7 |
| Protein (g) | 2.7 | 2.7 |
| Fibre (g) | 4.0 | 1.8 |
| Energy (kcal) | 36.6 | 41.4 |

*Concentrations expressed as mg of gallic acid equivalents per 12 g of cocoa powder.

heart rate monitor (Polar M430, Polar, Kempele, Finland); respiratory gas exchange measurements were taken using an automated gas analyser (Vyntus CPX Metabolic Cart, Vyaire Medical, Mettawa, IL, USA) to determine the rate of oxygen consumption ($\dot{V}_{O_2}$). The highest 30 s average of $\dot{V}_{O_2}$ was considered to be the $\dot{V}_{O_2peak}$. After finishing the test, the cycle ergometer workload was quickly decreased to 50 W for a 5 min cool-down.

### High- and low-flavanol interventions

Cocoa flavanol beverages were prepared by dissolving 12 g of cocoa powder into 350 ml of commercially available bottled 'Buxton' still natural mineral water. Buxton water was selected because of its low levels of nitroso species (nitrate: <0.1 mg/l), in order to avoid any external dietary influences on NO bioavailability (Lundberg & Govoni, 2004). During the study days, this was the only water provided to volunteers. The two cocoa powders used are commercially available (Barry Callebaut AG, Zurich, Switzerland). Specifically, the low-flavanol cocoa powder was a fat-reduced alkalized cocoa powder (commercial name: 10/12 DDP 'Royal Dutch') containing less than 6 mg of (−)-epicatechin and 5.6 mg of total flavanols per beverage. The high-flavanol cocoa powder was a fat-reduced natural cocoa powder (commercial name: Acticoa Natural) containing 150 mg of (−)-epicatechin and 695 mg of total flavanols per beverage. Levels of macro-nutrient content and breakdown of flavonoids are reported in Table 1.

The two cocoa-based flavanol interventions were matched for all micro- and macronutrients, including methylxanthines (caffeine and theobromine). Cocoa powder concentrations for flavanol monomers,

procyanidins and methylxanthines were measured by high-performance liquid chromatography, as described in previous studies (Alsolmei et al., 2019; Robbins et al., 2012). Total polyphenol concentrations were estimated by a Folin-Ciocalteu reagent calorimetric assay, as described previously (Miller et al., 2008). The dose of flavanol monomers used in the present study is in line with previous studies shown to be safe and effective in modifying human endothelial function acutely (Bapir et al., 2022; Heiss et al., 2003; Sansone et al., 2017; Schroeter et al., 2006) and plasma NO levels in healthy adults (Loke et al., 2008). Furthermore, similar monomer doses, such as (−)-epicatechin, can be achieved through diet by consuming a variety of flavanol-rich foods, such as green/black tea, unprocessed cocoa and berries (Bhagwat et al., 2014). The individual sachets of cocoa powder were labelled with an alphanumeric ID, and were stored at a temperature of −20°C. The two cocoa beverages were indistinguishable in texture, aroma and taste; they were provided in an opaque container (covered on top) with a dark-coloured opaque straw to maintain double-blindness. The unblinding process of the cocoa interventions was performed after completion of all data analyses.

## Flow-mediated dilatation of the brachial and superficial femoral arteries

Endothelial-dependent vasodilatation of the SFA and BA was assessed using the FMD protocol in agreement with established guidelines (Thijssen et al., 2019). Artery diameter and blood velocity of the SFA and BA were non-invasively assessed by means of a high-resolution duplex ultrasound device (Terason uSmart 3300, Teratech Corporation, Burlington, MA, USA) with a 15–4 MHz linear array transducer (Terason 15L4 Smart Mark, Teratech Corporation, Burlington, MA, USA), using a set frequency of 4.5 MHz. Pulse-wave Doppler signal was corrected at an insonation angle of 60°, and the sample volume (size: 1.5 mm) was placed at the centre of the artery lumen. The right SFA was located and scanned longitudinally 10–20 cm distal to the inguinal crease. A manual BP cuff was positioned around the distal end of the right thigh (3–4 cm proximal to the patella), distal to the imaged artery. We decided to perform the FMD protocol in the SFA, with the cuff placed around the thigh, because this was the only protocol (in the lower limbs) that had been demonstrated to be NO-mediated (Kooijman et al., 2008). The right BA was located and scanned longitudinally between 5 and 10 cm proximal to the antecubital fossa. A manual BP cuff (different from the cuff used during SFA FMD) was positioned around the right forearm (∼2 cm distal to the antecubital fossa),

distal to the imaged artery. In both the SFA and BA FMD protocols, once a satisfactory image of the artery was obtained (with defined vascular walls), the ultrasound probe was stabilised by means of an adjustable stereotactic probe-holding tool (FMD-probe-holder-xyz, Quipu S.r.l., Pisa, Italy). Artery diameter and blood velocity were continuously recorded for 1 min (baseline), 5 min in which the cuff was inflated and maintained at a pressure of 220 mm Hg, and 5 min following the rapid cuff deflation (<3 s). The FMD protocol duration was 11 min in total for each assessment. The location of the transducer on the participant's skin was marked and recorded to ensure consistency in placement during subsequent FMD measurements. All the FMD protocols were performed by a trained and experienced PhD student (AD, first author), with inter-day coefficients of variation for arterial diameter (mm) of 2.2% and 2.8% for BA and SFA, respectively, and for FMD (%), 10.9% and 8.3% for BA and SFA, respectively. These coefficients of variation are indicative of good/excellent reproducibility, and within the range suggested by FMD guidelines (Thijssen et al., 2019).

Measurements of artery diameter and blood velocity were analysed offline using an automated edge-detection software (Cardiovascular Suite, Quipu S.r.l., Pisa, Italy). All video recordings were analysed by the same researcher (AD) who performed the FMD measurements. Time-averaged maximum velocity (calculated by the software) was adjusted to reflect time-averaged mean velocity as described in previous research (Hoiland et al., 2015; Seidel et al., 1999) and using the following formula: *Time-Averaged Mean Velocity = Time-Averaged Maximum Velocity/2*. The resulting blood velocity was used to calculate arterial blood flow using the following formula: *Blood Flow = [Blood Velocity × π (Baseline Diameter/2)² ] × 60*. Baseline diameter was defined as the average diameter recorded during the minute preceding cuff inflation (i.e. the baseline). Peak diameter was defined as the largest diameter observed following cuff deflation. Both parameters were then used to calculate the percentage change in arterial diameter (i.e. FMD [%]) as described: *FMD (%) = [(Peak Diameter – Baseline Diameter)/Baseline Diameter] × 100*. Previous research indicates that the classic FMD (%) has a statistical bias towards baseline diameter. Atkinson and Batterham (2013a) proposed the use of allometrically scaled FMD (%) as an alternative method that minimises the influence of baseline diameter on FMD. In the present study, we have calculated allometrically scaled FMD (%) in accordance with their guidelines (Atkinson & Batterham, 2013b). Baseline values of shear rate, an adequate surrogate measure of shear stress (Pyke & Tschakovsky, 2005), was estimated using the formula: *Shear Rate = (4 × Baseline Blood Velocity)/Baseline Diameter*.

## Muscle microvasculature haemodynamics

Skeletal muscle microvascular haemodynamics were assessed in the leg using a NIRS device (NIRO-200NX, Hamamatsu Photonics KK, Shizuoka, Japan) during (1) the 2 h sitting trial, and (2) the SFA FMD protocol to assess microvasculature haemodynamics in response to upstream arterial occlusion. The NIRS probe – consisting of a light emitter and light detector (distanced 38 mm apart) – was placed on the skin of the right medial gastrocnemius and covered with an opaque cover to block the external ambient light, which may lead to measurement errors (Ghatas et al., 2020). Before applying the NIRS probe, the skin on the target area was shaved and cleaned for better adhesion of the probes. The probe was secured with breathable medical tape (3M Micropore Medical Tape, 3M United Kingdom PLC, Bracknell, England). The NIRS device can detect changes in the concentrations of oxyhaemoglobin and deoxyhaemoglobin, providing measures of tissue oxygen saturation and tissue haemoglobin concentration. Measures of tissue oxygenation index (TOI) and normalised tissue haemoglobin index (nTHI) (relative value of total haemoglobin content normalised to the initial value) were estimated via the spatially resolved spectroscopy method (Davies et al., 2015). The NIRS signals were acquired at a sample interval of 0.2 s (5 Hz). The medial gastrocnemius was selected for NIRS-related measures specifically because this muscle receives blood perfusion from branches of the SFA, which is scanned (and distally occluded) during SFA FMD.

**Reactive hyperaemia haemodynamics during SFA occlusion.** TOI (%) data were filtered using a 5 s moving average to minimise the influence of artefacts/noise in the data. As illustrated in Fig. 3: (1) baseline was determined as the average of 30 s during the resting period of the SFA FMD protocol; (2) minimum was calculated as the lowest value attained during ischaemia; (3) maximum was calculated as the highest value attained after cuff release; (4) $\Delta$TOI (%) was calculated as the difference between the baseline and the minimum; (5) overshoot was calculated as the difference between the maximum and the baseline; (6) reperfusion magnitude is the difference between the maximum and the minimum; (7) desaturation slope (%/s) is defined as the slope of the desaturation curve (the gradual TOI reduction during the 5 min ischaemia) starting at baseline and ending at the lowest point of the curve; (8) recovery slope (%/s) was quantified as the upslope of the TOI signal during a 10 s window following cuff deflation.

**Sitting trial.** Both TOI and nTHI were continuously measured during the 2 h sitting trial. For the purpose of the analysis, the following time points were considered

during sitting: start (0), 10 min, 60 min and end (120 min). For each time point, data were averaged over a 60 s period.

## Blood pressure and heart rate

Blood pressure and heart rate were measured in the left upper arm in supine position (following >10 min of supine resting) using an automatic BP monitor (Omron M3 [HEM-7131-E], OMRON HEALTHCARE Co., Ltd., Kyoto, Japan) to obtain systolic BP, diastolic BP and heart rate. Within each time point (e.g. pre-intervention), BP measurements were taken three consecutive times, which were then averaged.

## Statistical analysis

All statistical analyses were performed using statistical software IBM SPSS Statistics for Windows, version 29 (IBM Corp., Armonk, New York, USA). Independent sample *t* tests were performed to determine differences in participants' baseline characteristics between fitness groups (high fit *vs.* low fit). Three-way repeated measures ANOVA with sitting time (0; 2 h) and dietary intervention (low flavanol; high flavanol) as the within-subject variables, and fitness (high fit; low fit) as the between-subjects variable were performed to analyse changes in SFA and BA FMD, arterial diameter, shear rate, blood flow and BP. NIRS-related haemodynamic measures were also analysed using a similar approach. TOI and nTHI during sitting were analysed by a three-way repeated measures ANOVA with sitting time (0, 10, 60, 120 min), dietary intervention (low flavanol; high flavanol) and fitness (high fit; low fit) as the within/between subject variables. When statistical

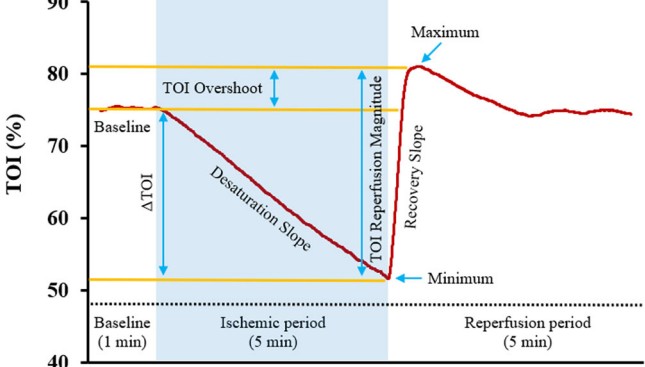

**Figure 3. Physiological response curve of TOI (%) for a representative participant measured during superficial femoral artery flow-mediated dilatation, including the near-infrared spectroscopy-related variables**
TOI, tissue oxygenation index.

**Table 2. Participants' baseline characteristics**

|  | High fit | Low fit | *F*-statistic and *P*-value |
|---|---|---|---|
| *N* | 20 | 20 |  |
| **Anthropometric and fitness** |  |  |  |
| Height (m) | 1.76 ± 0.06 | 1.79 ± 0.09 | *F*(1, 38) = 8.172, *P* = 0.245 |
| Weight (kg) | 70.4 ± 8.8* | 83.5 ± 11.9 | **F(1, 38) = 0.930, P < 0.001** |
| BMI (kg·m$^{-2}$) | 22.7 ± 2.5* | 26.0 ± 2.3 | **F(1, 38) = 0.002, P < 0.001** |
| Age (years) | 22.2 ± 2.9 | 23.2 ± 4.1 | *F*(1, 38) = 3.144, *P* = 0.354 |
| $\dot{V}_{O_2 peak}$ (ml·kg$^{-1}$·min$^{-1}$) | 56.4 ± 5.3* | 34.8 ± 4.6 | **F(1, 38) = 0.055, P < 0.001** |
| **Brachial artery** |  |  |  |
| Artery diameter (mm) | 4.0 ± 0.3 | 3.9 ± 0.4 | *F*(1, 38) = 3.598, *P* = 0.189 |
| Anterograde shear rate (s$^{-1}$) | 99.2 ± 42.5* | 148.4 ± 60.8 | **F(1, 38) = 0.985, P = 0.005** |
| Retrograde shear rate (s$^{-1}$) | −23.0 ± 11.6 | −24.1 ± 22.7 | *F*(1, 38) = 5.917, *P* = 0.852 |
| Anterograde blood flow (ml·min$^{-1}$) | 38.7 ± 21.0 | 51.6 ± 27.0 | *F*(1, 38) = 0.278, *P* = 0.099 |
| Retrograde blood flow (ml·min$^{-1}$) | −8.6 ± 4.6 | −8.1 ± 7.6 | *F*(1, 38) = 5.167, *P* = 0.772 |
| FMD (%) | 5.0 ± 2.3 | 6.2 ± 2.4 | *F*(1, 38) = 0.053, *P* = 0.119 |
| **Superficial femoral artery** |  |  |  |
| Artery diameter (mm) | 6.5 ± 0.5 | 6.4 ± 0.5 | *F*(1, 38) = 0.031, *P* = 0.324 |
| Anterograde shear rate (s$^{-1}$) | 88.7 ± 20.1* | 102.6 ± 22.1 | **F(1, 38) = 0.131, P = 0.043** |
| Retrograde shear rate (s$^{-1}$) | −28.4 ± 10.8 | −36.5 ± 15.3 | *F*(1, 38) = 6.233, *P* = 0.061 |
| Anterograde blood flow (ml·min$^{-1}$) | 142.5 ± 35.8 | 156.0 ± 41.8 | *F*(1, 38) = 1.020, *P* = 0.279 |
| Retrograde blood flow (ml·min$^{-1}$) | −45.1 ± 15.7 | −55.7 ± 24.1 | *F*(1, 38) = 9.616, *P* = 0.108 |
| FMD (%) | 3.8 ± 1.2 | 4.0 ± 1.1 | *F*(1, 38) = 0.510, *P* = 0.566 |
| **Blood pressure and heart rate** |  |  |  |
| Systolic BP (mm Hg) | 117.9 ± 9.1 | 121.3 ± 8.5 | *F*(1, 38)<0.001, *P* = 0.225 |
| Diastolic BP (mm Hg) | 63.2 ± 5.4* | 68.8 ± 4.5 | **F(1, 38) = 0.014, P = 0.001** |
| Heart rate (bpm) | 50.9 ± 7.4* | 63.1 ± 9.6 | **F(1, 38) = 0.223, P < 0.001** |

Data are presented as means ± SD.
*Denotes significant difference between fitness groups within the same parameter. BMI, body mass index; BP, blood pressure; FMD, flow-mediated dilatation; $\dot{V}_{O_2}$, rate of oxygen consumption. Data analysed via independent sample *t* tests.

significance was found for interaction effects, Bonferroni *post hoc* analysis was performed. Sample size estimation was performed using G*Power (version 3.1.9.3) and was based on acute changes in BA FMD following a similar dose of flavanol intake previously collected in our laboratory in young healthy males (Cohen's d = 0.91). For a standard deviation (SD) of 1.25% in this population (*P* < 0.05; power >85%), we estimated that 13 participants in each fitness group were needed to determine a significant within-subject difference between high- and low-flavanol interventions 2 h post-intake of at least 1.4% brachial FMD. Given that for SFA FMD (primary outcome measure) inter-day variability can be higher (Daniele et al., 2024) and render more frequently unusable data, we aimed to recruit *N* = 20 per group. The data in the tables and figures are presented as means ± SD. A *P*-value less than 0.05 was considered statistically significant. Non-usable data for SFA FMD (*N* = 9/40) were due to an inability to recover a satisfactory B-mode image post-occlusion in some of the time points. In regard to NIRS, the non-usable data (*N* = 4/40) were due to data files presenting high-amplitude, high-frequency artefacts, thus being considered unreliable and also reflect

occasions in which the NIRS device reported 'signal underflow', which reflects poor signal quality.

## Results

### Study participants

Participants' anthropometric and physiological characteristics are summarised in Table 2. All participants were young healthy males, aged 18–34 years old, with measures of body mass index (BMI), heart rate and BP all within a healthy range. The two fitness groups (high fit [HF]: *N* = 20; low fit [LF]: *N* = 20) shared similar height (HF: 1.76 ± 0.06 m; LF: 1.79 ± 0.09 m) and age (HF: 22.2 ± 2.9 yr.; LF: 23.2 ± 4.1 yr.). The high-fit group had lower weight and BMI than their low-fit counterparts. As intended, $\dot{V}_{O_2 peak}$ was higher in the high-fit group than the low-fit group (56.4 ± 5.3 *vs.* 34.8 ± 4.6 ml·kg$^{-1}$·min$^{-1}$; *P* < 0.001). Resting anterograde shear rate in both BA (*F*[1, 38] = 0.985, *P* = 0.005) and SFA (*F*[1, 38] = 0.131, *P* = 0.043) was lower in high-fit individuals. Diastolic BP (*F*[1, 38] = 0.014, *P* = 0.001) and heart rate (*F*[1, 38] = 0.223, *P* < 0.001) were also lower in high-fit than

**Table 3. Participants' daily sedentary time and physical activity**

| Daily physical activity pattern | High fit | Low fit | *F*-statistic and *P*-value |
|---|---|---|---|
| Sedentary activity time (h) | 10.4 ± 1.0 | 10.0 ± 2.2 | *F*(1, 30) = 9.195, *P* = 0.524 |
| Light activity time (h) | 1.4 ± 0.4 | 1.2 ± 0.4 | *F*(1, 30) = 0.354, *P* = 0.254 |
| Moderate activity time (h) | 2.8 ± 0.5 | 2.6 ± 0.8 | *F*(1, 30) = 3.012, *P* = 0.403 |
| Vigorous activity time (h) | 0.6 ± 0.4* | 0.1 ± 0.1 | ***F*(1, 30) = 17.572, *P* = 0.001** |
| Sedentary activity volume (estimated as MET·min) | 765.2 ± 80.3 | 716.3 ± 158.4 | *F*(1, 30) = 8.089, *P* = 0.271 |
| Light activity volume (estimated as MET·min) | 200.5 ± 52.7 | 176.5 ± 63.4 | *F*(1, 30) = 0.390, *P* = 0.250 |
| Moderate activity volume (estimated as MET·min) | 685.1 ± 146.5 | 622.5 ± 205.2 | *F*(1, 30) = 2.036, *P* = 0.324 |
| Vigorous activity volume (estimated as MET·min) | 336.0 ± 286.9* | 70.0 ± 75.2 | ***F*(1, 30) = 12.194, *P* = 0.002** |
| Step count | 12,421.7 ± 2,333.8* | 9,896.5 ± 3663.6 | ***F*(1, 30) = 2.430, *P* = 0.025** |

Recommendations for adults (aerobic physical activity, per week) – at least 150–300 min of moderate intensity, or at least 75–150 min of vigorous intensity (Bull et al., 2020). Data are presented as means ± SD.
*Denotes significant difference between fitness groups within the same parameter. MET, metabolic equivalent of task. High fit: *N* = 17; Low fit: *N* = 15. Data analysed via independent sample *t* tests.

low-fit individuals. All the other physiological measures were similar in both groups (Table 2).

## Habitual physical activity patterns

Participants' daily sedentary time and physical activity levels are reported in Table 3. Seven-day averaged accelerometry data showed that the high-fit group spent more time doing vigorous activity (HF: 0.6 ± 0.4 h; LF: 0.1 ± 0.1 h) and had a higher vigorous activity volume (HF: 336.0 ± 286.9 metabolic equivalent of task (MET)·min; LF: 70.0 ± 75.2 MET·min) in comparison to the low-fit group, but there were no differences in light and moderate physical activity levels between groups. The high-fit group had a higher step count (HF: 12,421.7 ± 2,333.8 steps; LF: 9,896.5 ± 3,663.6 steps). No significant differences were detected in sedentary time between fitness groups.

## Habitual dietary intake

Estimates of participant's daily dietary intake of macronutrients, total flavonoids and portions of fruit and vegetables are displayed in Table 4. The percentage of individuals who exceed or fail to meet the daily dietary recommendations (based on NHS guidance (NHS, 2022a, 2022b)) is also summarised (Table 4). The average daily intake of fibre was higher for the high-fit group compared with the low-fit group (16.7 ± 6.0 g *vs.* 12.1 ± 7.4 g, respectively), although most individuals failed to meet the recommended daily intake of fibre. Both fitness groups consumed similar amounts of flavonoids, and fruit and vegetables. In the high-fit group, only 35.3% were below the recommended daily intake of fruit and vegetables (portions) compared with 66.7% in the low-fit group.

## Resting shear rate, blood flow and arterial diameter

Sitting resulted in a decline in anterograde shear rate (Fig. 4*A* and *B*) and anterograde blood flow (Fig. 4*C* and *D*) in both the BA and SFA arteries ($P < 0.001$), regardless of flavanol intervention and fitness status. No differences in retrograde shear rate and blood flow were detected in either artery due to sitting, flavanol intervention or fitness (data not shown).

The low-fit group had higher anterograde shear rate in both the BA ($F[1, 38] = 8.555$, $P = 0.006$) and SFA ($F[1, 38] = 5.403$, $P = 0.026$), compared with the high-fit group, but there were no differences in anterograde blood flow between the fitness groups in both the BA ($F[1, 38] = 2.801$, $P = 0.102$) and SFA ($F[1, 38] = 1.777$, $P = 0.190$) (Fig. 4). Furthermore, the high-fit group experienced a smaller decline in anterograde shear rate in the BA ($P = 0.002$, $\Delta = -21.5 \text{ s}^{-1}$) during sitting compared with the low-fit group ($P < 0.001$, $\Delta = -44.3 \text{ s}^{-1}$), with fitness groups displaying significant differences post-sitting ($P = 0.012$). Flavanol interventions had no impact on anterograde/retrograde shear rate and blood flow. Sitting and fitness had no impact on arterial diameter in both BA and SFA. However, note that in the BA, there was a significant interaction effect (Flavanol × Fitness; $F(1, 38) = 5.315$, $P = 0.027$).

## Flow-mediated dilatation

Figure 5 shows BA and SFA FMD as well as allometrically scaled FMD before and after 2 h of sitting for both fitness groups after either a low- or high-flavanol acute intervention.

In both fitness groups, sitting resulted in a decline in BA FMD (HF: $\Delta = -0.6 \pm 1.2$%; LF: $\Delta = -0.7 \pm 1.0$%; Fig. 5*A*) and SFA FMD (HF: $\Delta = -1.0 \pm 0.6$%; LF: $\Delta = -1.1 \pm 1.1$%; Fig. 5*B*), but only following

**Table 4. Estimated daily dietary intake of key nutrients**

| Nutrients | Sample average | | % of participants over/under the recommended daily intake | | F-statistic and P-value |
|---|---|---|---|---|---|
| | High fit | Low fit | High fit | Low fit | |
| Energy (kcal) | 1935.9 ± 656.8 | 1851.8 ± 1024.3 | N/A | N/A | F(1, 33) = 0.324, P = 0.776 |
| Fat (g) | 77.2 ± 29.4 | 76.3 ± 48.8 | 58.8% over | 27.8% over | F(1, 33) = 0.366, P = 0.950 |
| Saturated fat (g) | 27.8 ± 10.6 | 29.4 ± 20.0 | 52.9% over | 33.3% over | F(1, 33) = 0.712, P = 0.767 |
| Carbohydrate (g) | 237.1 ± 89.4 | 211.5 ± 136.1 | N/A | N/A | F(1, 33) = 0.551, P = 0.518 |
| Sugars (g) | 98.3 ± 36.4 | 101.3 ± 75.1 | 100.0% over | 88.9% over | F(1, 33) = 3.870, P = 0.880 |
| Fibre (g) | 16.7 ± 6.0* | 12.1 ± 7.4 | 100.0% under | 94.4% under | **F(1, 33) = 0.004, P = 0.049** |
| Protein (g) | 81.1 ± 24.1 | 85.5 ± 28.8 | N/A | N/A | F(1, 33) < 0.001, P = 0.625 |
| Total flavonoids (mg) | 359.5 ± 263.0 | 304.4 ± 282.3 | N/A | N/A | F(1, 33) = 0.185, P = 0.555 |
| Fruit and vegetables (g) | 408.9 ± 167.1 | 325.7 ± 205.9 | N/A | N/A | F(1, 33) = 0.524, P = 0.200 |
| Fruit and vegetables (portion) | 5.1 ± 2.0 | 4.0 ± 2.5 | 35.3% under | 66.7% under | |

Recommendations for male adults – fat: <70 g/day, saturated fat: <30 g/day, sugar: <30 g/day, fibre: >30 g/day, fruit and vegetables: >5 portions/day. 1 portion = 80 g (NHS guidelines (NHS, 2022a)). Data are presented as means ± SD. *Denotes significant difference between fitness groups within the same nutrient. High fit: $N = 17$; Low fit: $N = 18$. Data analysed via independent sample $t$ tests.

consumption of the low-flavanol cocoa ($P < 0.001$), with no significant declines detected after the high-flavanol cocoa in both BA (HF: $\Delta = 0.2 \pm 1.2\%$; LF: $\Delta = 0.1 \pm 1.2\%$; $P = 0.403$) and SFA (HF: $\Delta = -0.2 \pm 0.9\%$; LF: $\Delta = 0.1 \pm 1.0\%$; $P = 0.679$). After sitting, there was a higher FMD following the high-flavanol compared with the low-flavanol for both the BA ($P = 0.010$) and SFA ($P < 0.001$) FMD measures. No significant differences between the flavanol interventions were detected at baseline in either FMD measures ($P = 0.111$ and $P = 0.846$, respectively).

Allometrically scaled FMD largely reflected what was observed for FMD. Both BA (Fig. 5C) and SFA (Fig. 5D) measures showed sitting-induced reductions following ingestion of the low-flavanol cocoa (BA: $P < 0.001$; SFA: $P < 0.001$), while no changes were observed in response to high-flavanol intake (BA: $P = 0.409$; SFA: $P = 0.573$). Similarly, higher FMDs were reported after sitting following the high-flavanol compared with the low-flavanol cocoa (BA: $P = 0.002$; SFA: $P < 0.001$), but no significant differences at baseline were observed (BA: $P = 0.243$; SFA: $P = 0.788$). Fitness did not influence the effects of sitting and flavanol intake on both FMD and allometrically scaled FMD.

**Leg muscle oxygenation haemodynamics during SFA occlusion**

Microvascular tissue oxygenation in the medial gastrocnemius during rest, SFA occlusion (ischaemic period), and post-occlusion reactive hyperaemia are depicted in Fig. 6A and B. Parameters describing the microvascular haemodynamic response (as represented in Fig. 3, Methods section) were estimated before and after sitting, following either a low- or high-flavanol intervention for both fitness groups (Table 5).

Sitting-induced declines in resting baseline oxygenation ($F[1, 34] = 145.110$, $P < 0.001$). After the release of the cuff (reperfusion period), sitting induced a reduction in maximum oxygenation ($F[1, 34] = 8.336$, $P = 0.007$), reperfusion magnitude (minimum–maximum; $F[1, 34] = 5.924$, $P = 0.020$) (Fig. 6C) and slower/reduced recovery slope ($F[1, 34] = 11.733$, $P = 0.002$; Fig. 6D). A significant increase in

**Table 5. Measures of tissue oxygenation index in the medial gastrocnemius during FMD in the superficial femoral artery**

| TOI variable | High fit | | | | Low fit | | | | Effect |
|---|---|---|---|---|---|---|---|---|---|
| | Low flavanol | | High flavanol | | Low flavanol | | High flavanol | | |
| | Pre | Post | Pre | Post | Pre | Post | Pre | Post | |
| Baseline (%) | 70.8 ± 3.7 | 66.7 ± 3.6 | 69.5 ± 5.5 | 66.1 ± 5.5 | 76.0 ± 3.4 | 71.6 ± 3.8 | 76.1 ± 4.2 | 71.5 ± 3.6 | Sitting: $F[1, 34] = 145.110$, $P < 0.001$; Fitness: $F[1, 34] = 20.074$, $P < 0.001$ |
| Minimum (%) | 44.9 ± 8.7 | 47.4 ± 6.7 | 45.4 ± 10.8 | 46.0 ± 7.3 | 53.5 ± 10.4 | 52.4 ± 8.3 | 52.7 ± 9.8 | 53.3 ± 8.8 | Fitness: $F[1, 34] = 6.915$, $P = 0.013$ |
| Maximum (%) | 77.1 ± 3.8 | 75.9 ± 4.5 | 76.1 ± 5.3 | 75.6 ± 5.4 | 80.7 ± 2.8 | 79.7 ± 3.5 | 81.4 ± 3.4 | 80.3 ± 3.9 | Sitting: $F[1, 34] = 8.336$, $P = 0.007$; Fitness: $F[1, 34] = 11.898$, $P = 0.002$ |
| ΔTOI (%) | 26.0 ± 8.3 | 19.3 ± 5.6*** | 24.1 ± 7.2 | 20.0 ± 5.4*** | 22.5 ± 9.2 | 19.2 ± 6.8* | 23.4 ± 8.7 | 18.2 ± 6.9*** | Sitting: $F[1, 34] = 49.154$, $P < 0.001$; Flavanol × Sitting × Fitness: $F[1,34] = 4.251$, $P = 0.047$ |
| Overshoot (%) | 6.3 ± 2.1 | 9.2 ± 3.1 | 6.6 ± 3.2 | 9.5 ± 3.5 | 4.8 ± 2.8 | 8.1 ± 3.0 | 5.2 ± 3.4 | 8.8 ± 4.2 | Sitting: $F[1, 34] = 124.398$, $P < 0.001$ |
| Reperfusion magnitude (%) | 32.2 ± 9.7 | 28.5 ± 7.7 | 30.7 ± 10.0 | 29.5 ± 7.8 | 27.3 ± 11.2 | 27.3 ± 9.2 | 28.6 ± 11.0 | 27.0 ± 10.5 | Sitting: $F[1, 34] = 5.924$, $P = 0.020$ |
| Desaturation slope (%/s) | −0.09 ± 0.03 | −0.06 ± 0.02*** | −0.08 ± 0.02 | −0.07 ± 0.02*** | −0.08 ± 0.03 | −0.06 ± 0.02* | −0.08 ± 0.03 | −0.06 ± 0.02*** | Sitting: $F[1, 34] = 49.192$, $P < 0.001$; Flavanol × Sitting × Fitness: $F(1, 34) = 4.263$, $P = 0.047$ |
| Recovery slope (%/s) | 1.05 ± 0.68 | 0.72 ± 0.37 | 0.92 ± 0.45 | 0.80 ± 0.41 | 0.92 ± 0.42 | 0.83 ± 0.45 | 1.01 ± 0.65 | 0.88 ± 0.62 | Sitting: $F[1, 34] = 11.733$, $P = 0.002$ |

Data are presented as means ± SD.
*** Denotes significant difference ($P < 0.001$) *vs.* pre-sitting.
* Denotes significant difference *vs.* pre-sitting (ΔTOI: $P = 0.025$, TOI desaturation slope: $P = 0.025$). TOI, tissue oxygenation index. High fit: $N = 19$; Low fit: $N = 17$. Statistical details ($F$, df, $P$) were presented exclusively for statistically significant findings. Three-way repeated measures ANOVA conducted. Bonferroni *post hoc* for significant interactions.

overshoot TOI (baseline–maximum; $F[1, 34] = 124.398$, $P < 0.001$) was observed after sitting. During the ischaemic period, sitting resulted in a reduction in $\Delta$TOI (baseline–minimum; $F[1, 34] = 49.154$, $P < 0.001$; Fig. 6E) and slower/reduced desaturation slope ($F[1, 34] = 49.192$, $P < 0.001$; Fig. 6F), but no changes in the minimum oxygenation reached during occlusion (Table 5).

Fitness did not affect the oxygenation haemodynamics during occlusion or reperfusion. There was generally lower oxygenation in high-fit individuals compared with low-fit, reflected in lower values at baseline ($F[1, 34] = 20.074$, $P < 0.001$), minimum ($F[1, 34] = 6.915$, $P = 0.013$), and maximum ($F[1, 34] = 11.898$, $P = 0.002$).

High- and low-fit individuals experienced a similar micro-vascular response to sitting.

In the low-fit group only, the low-flavanol intervention experienced a smaller drop ($P = 0.025$) in $\Delta$TOI ($\Delta$TOI = 3.3%) due to sitting compared to the high-flavanol ($\Delta$TOI = 5.5%). Furthermore, in the low-fit group only, those individuals who consumed flavanols experienced a lower decline in the rate of desaturation due to sitting compared with the low-flavanol group ($\Delta$desaturation slope = 0.01%/s, and $\Delta$ = 0.03%/s, respectively; $P < 0.001$). However, no differences between high and low flavanol were detected 2 h post-sitting for any parameter. Flavanols did not affect the haemodynamics of the reperfusion period.

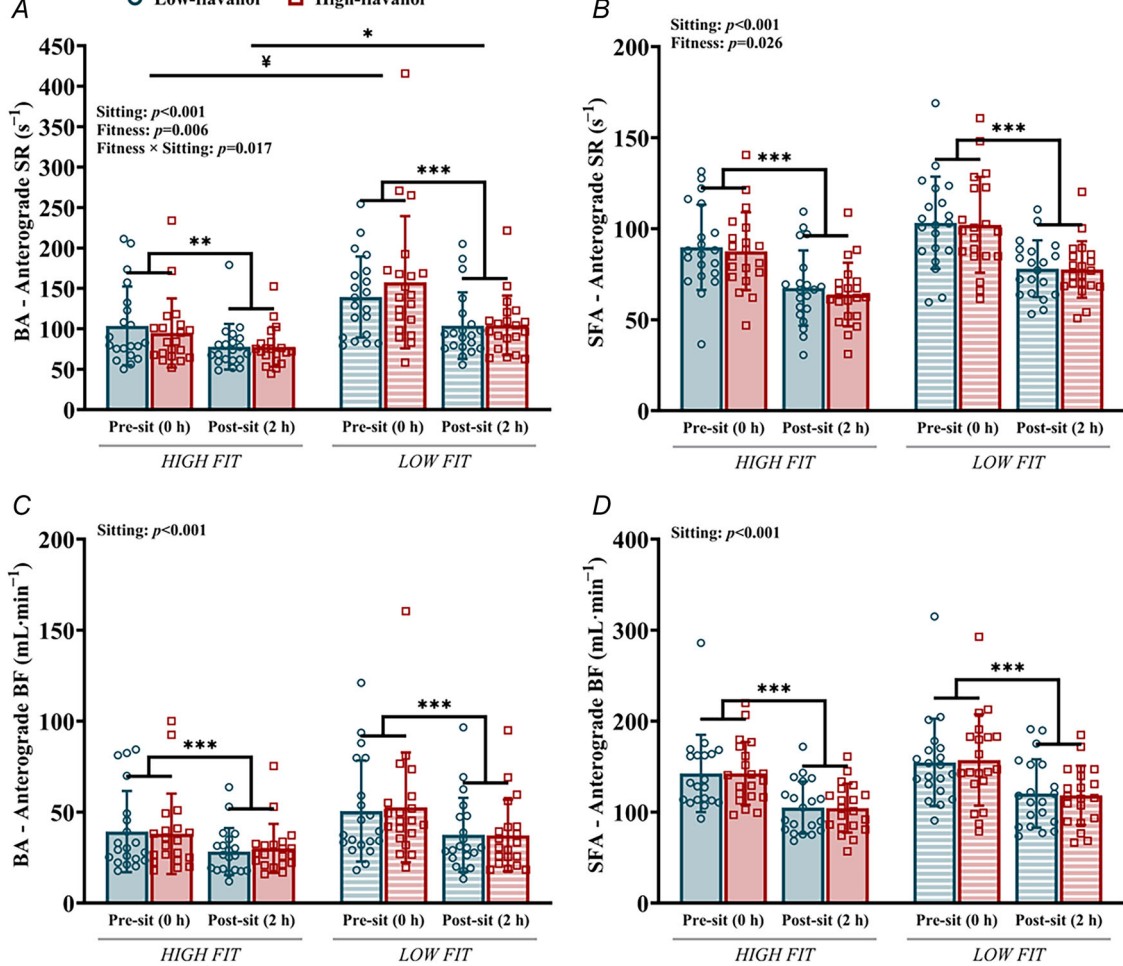

**Figure 4. Resting anterograde shear rate and blood flow of the brachial and superficial femoral arteries**
Resting anterograde shear rate and blood flow of the brachial *A* and *C*, and superficial femoral arteries *B* and *D*, in high- and low-fit participants, before (0 h) and after 2 h of sitting, following either a low- or high-flavanol intervention. Data are presented as means ± SD. **Denotes significant difference ($P = 0.002$) between pre- and post-sitting. ***Denotes significant difference ($P < 0.001$) between pre- and post-sitting. *Denotes significant difference ($P = 0.012$) between fitness groups post-sitting. ¥Denotes significant difference ($P = 0.005$) between fitness groups before sitting. BA, brachial artery; BF, blood flow; SFA, superficial femoral artery; SR, shear rate. Three-way repeated measures ANOVA conducted. Bonferroni *post hoc* for significant interactions.

## Leg muscle oxygenation haemodynamics during sitting

Microvascular tissue oxygenation (as reflected by TOI) and total haemoglobin content (as reflected by nTHI) in the medial gastrocnemius during the 2 h sitting trial are reported in Fig. 7.

Sitting induced a decline in TOI following the low-flavanol cocoa intake at 60 min ($P < 0.001$) and at 120 min ($P < 0.001$), while following the high-flavanol cocoa intake the declines were observed at all time points ($P < 0.001$). Sitting induced significant increases in nTHI in both fitness groups, with the high-fit group displaying significantly higher nTHI only at 10 min ($P = 0.009$), while the low-fit group significantly increased at 10 min ($P = 0.010$), 60 min ($P < 0.001$), and at 120 min ($P = 0.014$). No effects of flavanols were detected in nTHI during sitting.

There was a significant difference in TOI between the two fitness groups, with the high-fit group displaying lower TOI values ($F[1, 36] = 10.930$, $P = 0.002$). In regard to nTHI, the two fitness groups exhibited a significant difference at baseline (higher in the high-fit group [$P = 0.006$]). In addition, fitness played a role in modulating sitting-induced nTHI: in the high-fitness group, there was a significant increase in nTHI from start (0) to 10 min ($P = 0.009$) and a decline from 10 min to 120 min ($P = 0.014$), while in the low-fitness group, nTHI

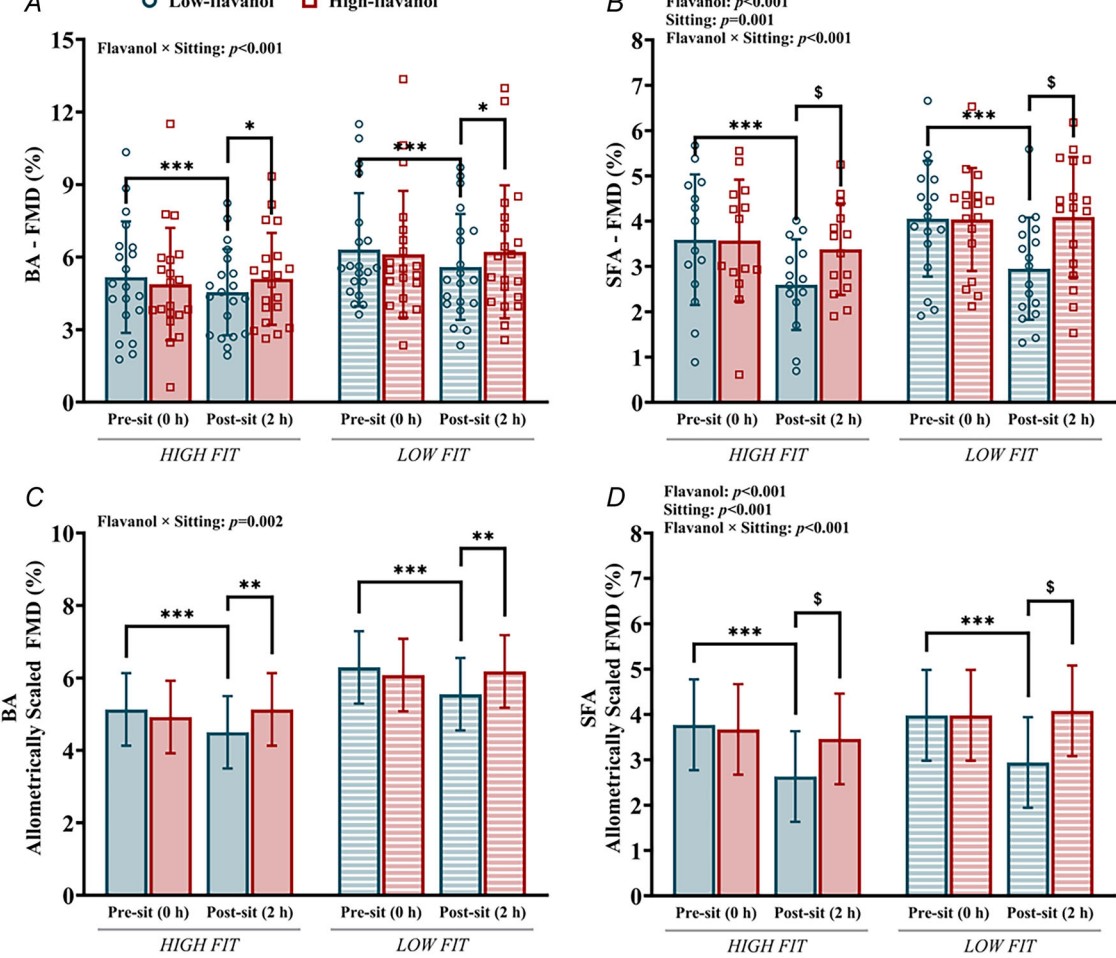

**Figure 5. Endothelial function of the brachial and superficial femoral arteries**
Endothelial function, measured as flow-mediated dilatation (FMD, %) of the brachial *A*, and superficial femoral artery *B*, and allometrically scaled FMD of the brachial *C*, and superficial femoral artery *D*, before (0 h) and after 2 h of sitting, following either a low- or high-flavanol intervention. Data are presented as means ± SD. ***Denotes a significant difference ($P < 0.001$) between pre- and post-sitting. **Denotes significant difference ($P = 0.002$) between the two flavanol interventions post-sitting. *Denotes significant difference ($P = 0.010$) between the two flavanol interventions post-sitting. $Denotes significant difference ($P < 0.001$) between the flavanol interventions post-sitting. BA, brachial artery; FMD, flow-mediated dilatation; SFA, superficial femoral artery. Superficial femoral artery (endothelial function): high fit, $N = 14$; low fit, $N = 17$. Three-way repeated measures ANOVA conducted. Bonferroni *post hoc* for significant interactions.

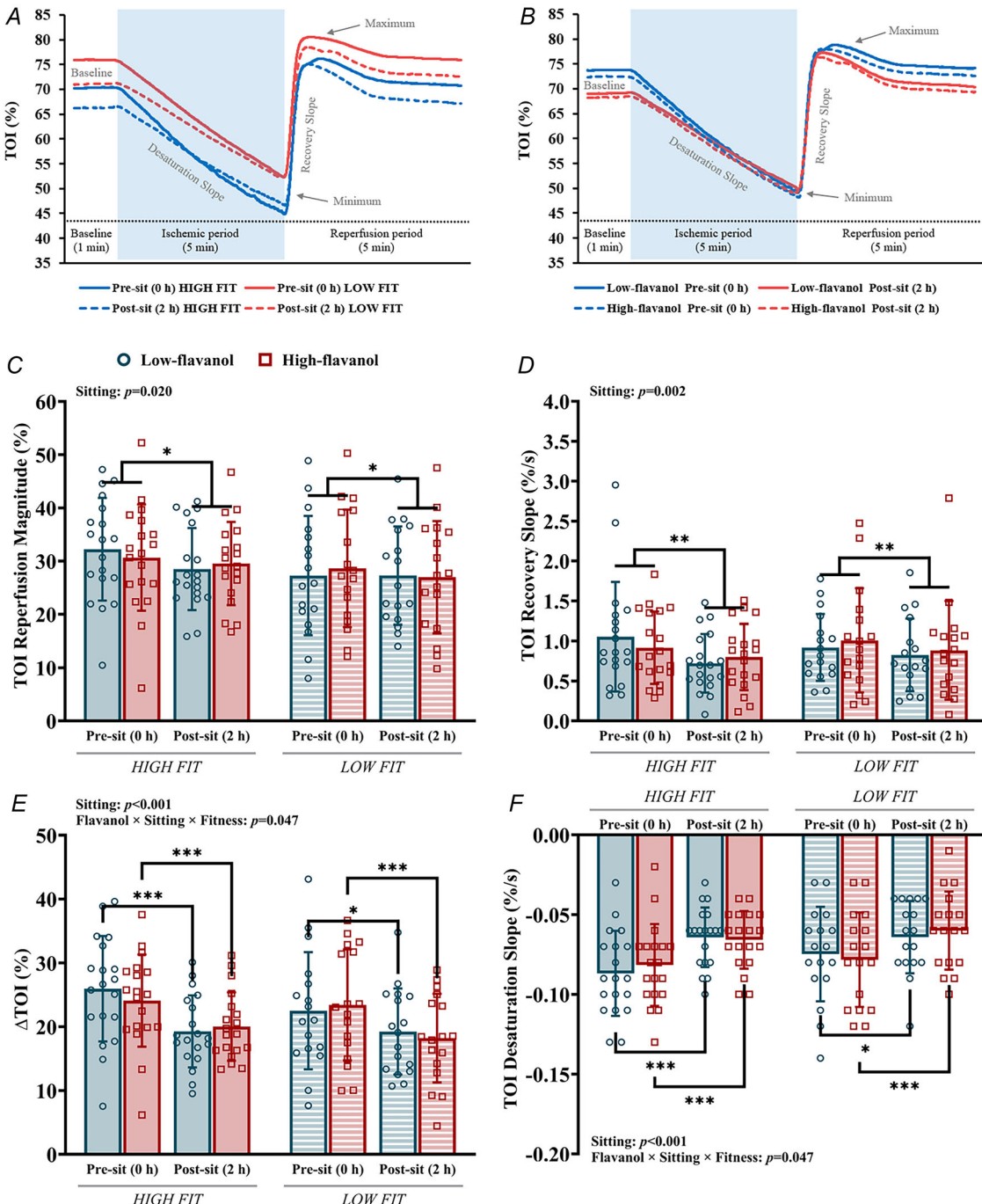

**Figure 6. Microvascular function measures in the medial gastrocnemius during reactive hyperaemia**
Averaged physiological response curves of TOI (%) measured during 5 min SFA occlusion (FMD) *A*, before and after 2 h of sitting in high (*N* = 19) and low (*N* = 17) fitness groups; *B*, before and after 2 h of sitting following either a high- or low-flavanol cocoa. This is followed by microvascular function measures of TOI in the medial gastrocnemius during SFA FMD. *C*, TOI reperfusion magnitude (%); *D*, TOI recovery slope (%/s); *E*, ΔTOI (%); *F*, TOI desaturation slope (%/s). Data are presented as means ± SD. *Denotes significant difference between pre- and post-sitting (TOI reperfusion magnitude: *P* = 0.020; ΔTOI: *P* = 0.025; TOI desaturation slope: *P* = 0.025). **Denotes significant difference (*P* = 0.002) between pre- and post-sitting. ***Denotes significant difference (*P* < 0.001) between pre- and post-sitting. TOI, tissue oxygenation index. Three-way repeated measures ANOVA conducted. Bonferroni *post hoc* for significant interactions.

was maintained higher across all time points during sitting ($P < 0.05$).

### Blood pressure and heart rate following sitting

No changes in systolic BP were detected after sitting ($F[1, 38] = 0.289$, $P = 0.594$) (Fig. 8A), while diastolic BP increased in both fitness groups ($F[1, 38] = 27.021$, $P < 0.001$), regardless of whether it was the low-flavanol (HF: $\Delta = 5.8 \pm 5.6$ mm Hg; LF: $\Delta = 2.7 \pm 5.6$ mm Hg) or high-flavanol intervention (HF: $\Delta = 5.6 \pm 6.6$ mm Hg; LF: $\Delta = 2.4 \pm 5.6$ mm Hg) (Fig. 8B). Heart rate decreased after the sitting trial ($F[1, 38] = 16.142$, $P < 0.001$), regardless of whether it was the low-flavanol (HF: $\Delta = -2 \pm 4$ bpm; LF: $\Delta = -5 \pm 10$ bpm) or high-flavanol intervention (HF: $\Delta = -3 \pm 6$ bpm; LF: $\Delta = -5 \pm 7$ bpm; Fig. 8C). Low-fit individuals displayed higher diastolic BP ($F[1, 38] = 5.897$, $P = 0.020$) and heart rate ($F[1,$

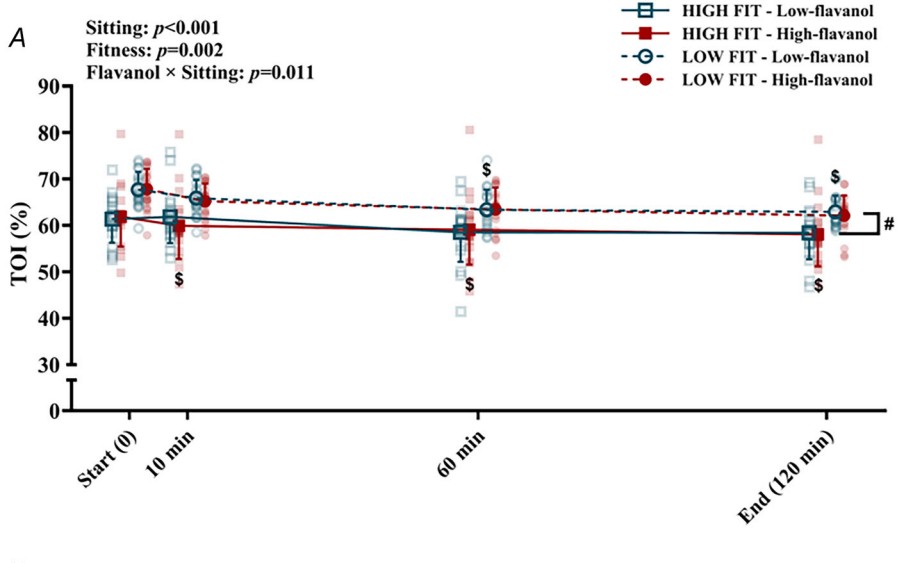

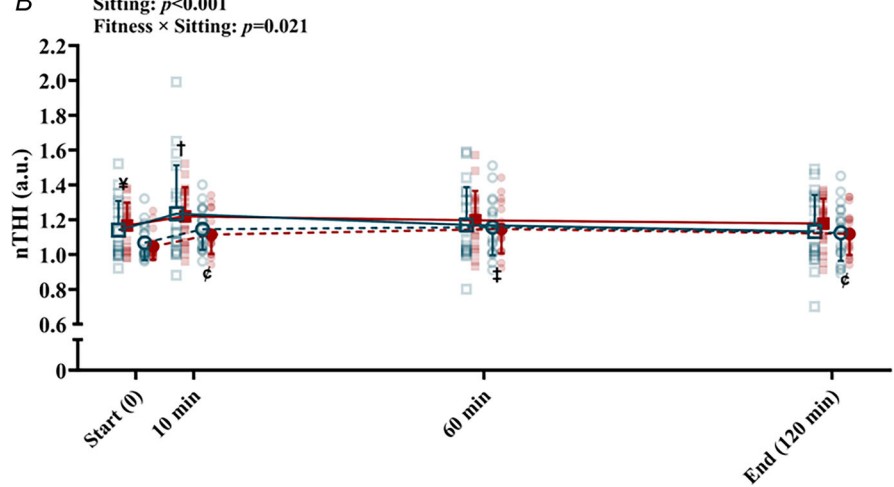

**Figure 7. Microvascular function measures in the medial gastrocnemius during 2 h of sitting in high (*N* = 19) and low (*N* = 19) fitness groups**

*A*, TOI (%); *B*, nTHI (a.u.). Data are presented as means ± SD. $^\$$Denotes significant difference ($P < 0.001$) between the post-intervention time point and 'Start (0)', within the same flavanol intervention. $^\#$Denotes significant difference ($P = 0.002$) between the two fitness groups. $^¥$Denotes significant difference ($P = 0.006$) between the two fitness groups at 'Start (0)'. $^¢$Denotes significant difference in both flavanols between the post-intervention time point (10 min: $P = 0.010$; 120 min: $P = 0.014$) and 'Start (0)', within the same fitness group. $^†$Denotes significant difference ($P = 0.009$) in both flavanol interventions between the post-intervention time point and 'Start (0)', within the same fitness group. $^‡$Denotes significant difference ($P < 0.001$) in both flavanol interventions between the post-intervention time point and 'Start (0)', within the same fitness group. nTHI, normalised tissue haemoglobin index; TOI, tissue oxygenation index. Three-way repeated measures ANOVA conducted. Bonferroni *post hoc* for significant interactions.

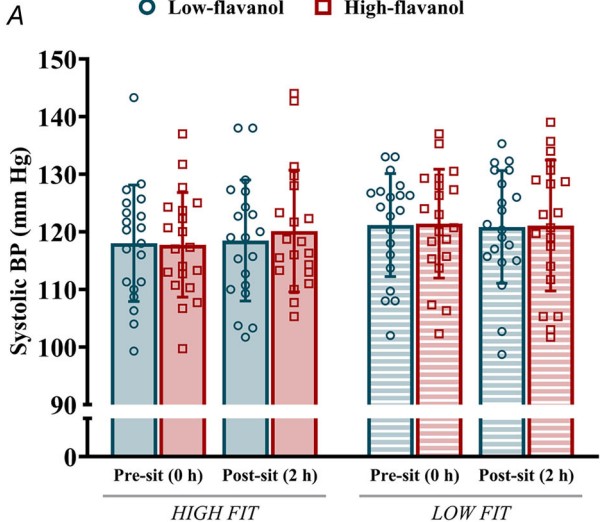

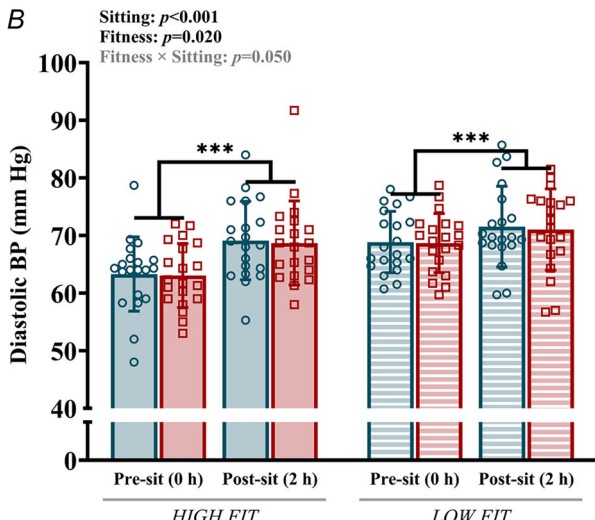

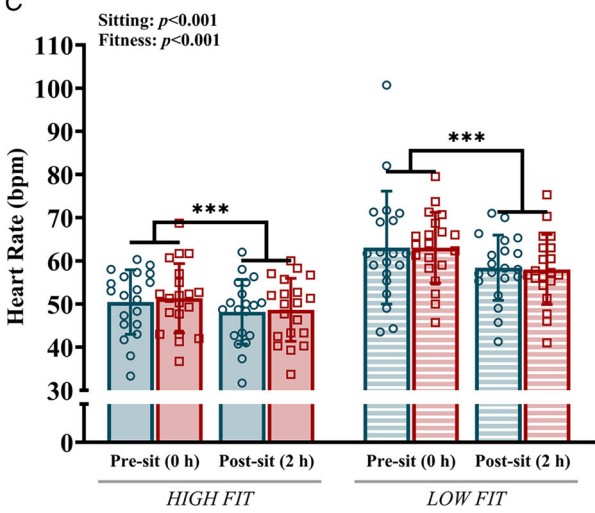

**Figure 8. Resting brachial blood pressure and heart rate**
Systolic *A*, and diastolic blood pressure *B*, and heart rate *C*, in high- and low-fit participants, before (0 h) and after 2 h of sitting,

following either a low- or high-flavanol intervention. Data are presented as means ± SD. ***Denotes a significant difference ($P <$ 0.001) between pre- and post-sitting. BP, blood pressure. Three-way repeated measures ANOVA conducted. Bonferroni *post hoc* for significant interactions.

38] = 22.546, $P < 0.001$) than their high-fit counterparts, but no differences in systolic BP ($F[1, 38] = 0.825$, $P = 0.369$).

## Discussion

This study aimed to investigate whether cocoa flavanols could be used as a dietary strategy to protect endothelial function and improve peripheral muscle oxygenation during sitting in young healthy male adults with different fitness levels. In addition, the study assessed whether higher fitness might reduce the detrimental effects of sitting on vascular function. The main findings were that 2 h of uninterrupted sitting induced declines in upper- and lower-limb endothelial function (as assessed by brachial and femoral FMD, respectively), anterograde shear rate and blood flow, as well as blunted peripheral muscle oxygenation responses and increased diastolic BP. This was observed in both high- and low-fit individuals, suggesting that higher levels of fitness are not protective against the detrimental effects of acute bouts of sitting. Dietary flavanols were capable of preventing sitting-induced reductions in both BA and SFA FMD, without further impacting blood flow, BP or muscle microvascular function. Furthermore, we have shown for the first time that baseline levels of cardiorespiratory fitness do not modulate the vascular effects of flavanol intake, indicating that individuals can benefit from flavanol intake regardless of their levels of physical fitness. To our knowledge, this is the first study to demonstrate the protective effects of dietary flavanols during sitting in young healthy men and their interactions with physical fitness.

### Macrovasculature: impact of sitting and fitness

In the present study, sitting-induced reductions in FMD in the BA were 0.7% and approximately 1.0–1.1% in the SFA in both fitness groups, which is less than what has been reported previously (i.e. reductions in SFA between 1.5% and 3.2% after 2 h of sitting in young healthy adults (Ballard et al., 2017; Caldwell et al., 2020; Thosar et al., 2014; Thosar, Bielko, Mather, et al., 2015; Thosar, Bielko, Wiggins, et al., 2015)). The differences in the magnitude of effect between studies may be attributable to differences in sample size, with the current study having a larger sample size than previously tested ($N = 11$–26 *vs.* 40). The

evidence for effects of sitting in the BA is more limited, but studies have generally reported no effect on FMD (Carter & Gladwell, 2017; Restaino et al., 2015; Thosar et al., 2014), with the exception of one study reporting FMD declines of up to 4% (Headid 3rd et al., 2020).

Consistent with the literature, we observed declines in anterograde shear rate and blood flow across upper- and lower-limb arteries (Ballard et al., 2017; Restaino et al., 2015; Thosar et al., 2014; Thosar, Bielko, Mather, et al., 2015; Thosar, Bielko, Wiggins, et al., 2015), but no significant changes in retrograde shear rate (Thosar et al., 2014; Thosar, Bielko, Mather, et al., 2015; Thosar, Bielko, Wiggins, et al., 2015). Alterations in anterograde shear rate/blood flow are thought to at least partially underlie changes in FMD (Holder et al., 2019; Tremblay et al., 2019), but whether that is the case in the context of sitting is currently unclear. Furthermore, a recent study demonstrated that prolonged sitting also induced impairments in endothelium-independent dilatation (nitroglycerin-mediated), along with endothelium-dependent dilatation (FMD) in the lower limbs, indicating that both reduced vascular smooth muscle sensitivity to NO and NO bioavailability contribute to sitting-induced declines in FMD (Liu et al., 2023).

Importantly, data from the present study indicate that such declines in blood flow/shear rate are similar in both high- and low-fit individuals, despite significant differences in baseline arterial shear rate between the groups. In line with this, both fitness groups show similar declines in FMD during sitting across the vasculature, suggesting that higher levels of cardiorespiratory fitness do not protect against sitting-induced vascular dysfunction (Garten et al., 2019; Liu et al., 2021). Previous studies report similar reductions in the popliteal artery FMD between fitness groups during sitting (Liu et al., 2021), and also impaired leg vascular function, as measured by passive leg movement (Garten et al., 2019). However, the current data are the first to show that NO-dependent endothelial function is affected by sitting irrespective of fitness, given that the SFA FMD protocol used in the present study has been demonstrated to largely rely on NO availability (Kooijman et al., 2008).

## Macrovasculature: impact of flavanol intake during sitting

Herein, observed for the first time, one dose of dietary flavanols proved to be effective at preventing sitting-induced vascular dysfunction in both upper and lower limbs in young healthy men. A recent systematic review and meta-analysis of 24 randomised controlled trials revealed that acute flavanol ingestion induces an increase in FMD of 1.7% (Raman et al., 2019). In the context of sitting, this effect appears to be of smaller magnitude (0.6% in BA FMD, and 1% in SFA FMD, in relation to placebo). It is important to note that flavanols were capable of maintaining pre-sitting FMD levels rather than improving FMD above baseline – as previously demonstrated (Rodriguez-Mateos et al., 2015; Sansone et al., 2017; Schroeter et al., 2006), which is probably a reflection of the negative impact of sitting on the vasculature. We have also demonstrated that such beneficial effects during sitting extend to the lower-limb vasculature in young healthy men. To our knowledge, there is only one other study that has shown beneficial effects of cocoa flavanols on lower-limb artery FMD (i.e. common femoral artery), in adults over 50 years old, both healthy and with type 2 diabetes (Bapir et al., 2022). Importantly, our data add to the literature by showing for the first time a similar protection of flavanols in both high- and low-fit young individuals, as well as in a NO-mediated conduit artery in the lower limb (Kooijman et al., 2008).

The improvements in endothelial function after flavanol intake are significant, but generally of lower magnitude when compared with some of the other physical activity-based interventions described previously in the literature. For example, treadmill exercising prior to sitting improves SFA FMD by 1.6% (Ballard et al., 2017). Additionally, 5 min walking breaks or stair climbing every hour can induce FMD improvements of 4.5% and 2.5% for SFA and BA, respectively (Cho et al., 2020; Thosar, Bielko, Mather, et al., 2015). This is perhaps to be expected given the nature of the two types of intervention: nutritional *vs.* physical activity-based. The latter is undoubtedly the ideal strategy to prevent/attenuate sitting-induced vascular dysfunction, but nutritional approaches might be used in combination with physical activity interventions or as an alternative, when activity-based strategies are not feasible or possible.

Interestingly, the benefits for endothelial function observed by breaking up sitting with standing or physical activity are thought to be at least partially mediated by elevations in shear rate and blood flow (Ballard et al., 2017; Caldwell et al., 2020; Morishima et al., 2016, 2017; Restaino et al., 2015, 2016). In contrast, cocoa flavanols improved FMD without affecting resting shear rate and blood flow, suggesting that a separate mechanism is at play. This is consistent with some of our previous studies (Baynham et al., 2021, 2024). Indeed, accumulating evidence indicates that cocoa flavanols, particularly (−)-epicatechin, upregulate endothelial NO synthase expression to increase the bioavailability of NO (Jaramillo Flores, 2019; Moreno-Ulloa et al., 2014; Schroeter et al., 2006), which plays a substantial role in mediating the FMD response (Green et al., 2014). Further, it has been shown that both acute and chronic intake of cocoa flavanols have no effect on endothelium-independent

vasodilatation (Rassaf et al., 2016), suggesting that cocoa flavanols do not affect vascular smooth muscle sensitivity to NO. Interestingly, and similar to the present findings, a recent study showed that FMD in the popliteal artery is maintained during sitting in endurance-trained individuals, despite a reduction in shear rate (Morishima et al., 2020), which indicates that other mechanisms (other than blood flow/shear rate maintenance) can play a role in the prevention of declines in endothelial function during sitting.

### Microvasculature: impact of sitting and fitness

Within the downstream muscle microvasculature, robust effects of sitting on tissue oxygenation responses were observed, which were not affected by fitness. Specifically, there were consistent declines in resting levels of tissue oxygenation during sitting, supported by both the continuous TOI measurements during the 2 h sitting period and by lower SFA blood flow and shear rate observed in the present study after sitting. This generally reflects previously published muscle oxygenation data during sitting (Evans et al., 2019; Stoner et al., 2019), suggesting an imbalance between oxygen supply and use, which may occur as a result of attenuated femoral endothelial function. During the occlusion period, sitting does not affect the extent of the ischaemia (minimum), but it slows down rates of desaturation, reflecting reduced muscle oxygen consumption, which is in line with previous work (Horiuchi & Stoner, 2022; Tamiya, Hotta, et al., 2024). During the hyperaemic period, sitting reduces the magnitude of oxygen reperfusion (also reflected in a smaller maximum TOI) and further reduces the speed of recovery (recovery slope), indicating that after sitting, efficiency of re-oxygenation may be reduced. These data are in line with previous sitting (2.5–3 h) studies of the medial gastrocnemius or soleus muscle (Anderson & Park, 2023; Headid 3rd et al., 2020; Horiuchi & Stoner, 2022; Park et al., 2022; Tamiya, Hotta, et al., 2024), and generally reflects a transient microvascular dysfunction in the leg muscles in response to prolonged sitting.

Interestingly, high-fit individuals did not experience better microvasculature function following sitting than the low-fit individuals, suggesting that higher fitness may not be protective. Previous literature suggests that high-fit individuals are more efficient at oxygen extraction (Kalliokoski et al., 2001; Montero et al., 2015; Skattebo et al., 2020; Tuesta et al., 2024), which is in line with the lower resting levels of TOI observed in the current data and by others (Yogev et al., 2023). During sitting, while both high- and low-fit groups experienced a similar decline in TOI, the low-fit group increased nTHI during the 2 h period, reaching levels of nTHI similar to those of the high-fit group, by the end of the sitting period. This

may suggest a compensatory mechanism in the low-fit group to deal with the decline in oxygenation during sitting. From a functional point of view (during reactive hyperaemia), no differences in magnitude of reperfusion or recovery post-occlusion were detected between the high- and low-fit groups. Previous findings report that trained individuals display a steeper recovery slope than their untrained counterparts (George et al., 2018; McLay et al., 2016; Soares et al., 2018).

### Microvasculature: impact of flavanol intake during sitting

Flavanols had no effect on microvascular function (post-hyperaemia response) post-sitting. The high flavanol reduced the rate of desaturation after sitting in the low-fitness group only, suggesting that flavanols may mask inactivity-driven muscle oxygen consumption, but the relevance of this finding is unclear. Previous work assessing the acute effects of cocoa flavanols in reactive hyperaemia in the forearm skeletal muscle using NIRS reported no significant effects on TOI desaturation rate, TOI recovery rate or TOI reperfusion magnitude (Santos et al., 2023). On the other hand, studies assessing microvascular function (cutaneous or muscular) using other methods have reported positive effects of cocoa flavanols (Baynham et al., 2021; Heiss et al., 2015; Neukam et al., 2007), reflecting contrasting results in the literature (Richardson et al., 2025). The differential effects of flavanols across the macro- and microvasculature (Bapir et al., 2022; Latif et al., 2024) might be related to the importance/contribution of NO in such processes. For example, the contribution of NO to peak reactive hyperaemia in the forearm and cutaneous microvasculature is thought to be limited, with more contribution arising from other vasoactive compounds, including prostaglandins and endothelium-derived hyperpolarising factors (Crecelius et al., 2013; Rosenberry & Nelson, 2020).

### Blood pressure

Sitting induced significant increases in diastolic BP in both high- and low-fit individuals, following both the low-flavanol and high-flavanol intervention, whereas systolic BP remained unchanged (see Fig. 8). Sitting-induced increases in diastolic BP (here up to 5.8 mm Hg) can reach pre-hypertensive BP ranges (Mancia et al., 2023) and may be of clinical significance, given that in young adults (age, $\leq 50$ years) diastolic BP can be a better predictor of mortality than systolic BP (Taylor et al., 2011). In agreement with the current data, several experimental studies (e.g. Morishima et al., 2020; O'Brien et al., 2019; Tamiya, Hoshiai, et al., 2024),

but not all (e.g. Carter et al., 2019; Headid 3rd et al., 2020; Restaino et al., 2016), have shown that acute exposure to prolonged sitting significantly increases BP (either systolic BP, diastolic BP, or mean arterial pressure) in young adults. A recent systematic review and meta-analysis has reported that uninterrupted prolonged sitting leads to small significant increases in both systolic BP (weighted mean difference = ~5 mm Hg) and mean arterial pressure (weighted mean difference = ~3 mm Hg) but diastolic BP was unaffected (Paterson et al., 2022). It has been proposed that sitting-induced increases in BP may be driven by impairments in peripheral endothelial function (Drożdż et al., 2023). Importantly, the present data indicate for the first time that a higher level of fitness is not protective against acute rises in BP due to sitting.

Furthermore, flavanols did not prevent sitting-induced increases in BP in high- or low-fit individuals. Short-term consumption of cocoa flavanols (2 to 18 weeks) has been shown to improve BP in healthy individuals (Amoah et al., 2022; Ried et al., 2012) but the acute benefits (within hours) have been more consistently shown in middle-aged/older populations, with limited research in young adults (Heiss et al., 2015; Sansone et al., 2017; Vlachopoulos et al., 2005). However, the acute studies in young adults report benefits in reducing systolic BP, but not diastolic BP (Heiss et al., 2015; Sansone et al., 2017), which may explain why acute supplementation was not effective in the context of sitting. In summary, findings from the present study indicate that flavanols can prevent endothelial dysfunction, but do not improve BP during sitting. Longer periods of supplementation may be needed to build resilience against rises in BP during sedentary periods.

### Habitual diet and physical activity

In the present study, there are no significant differences in intake of flavonoids, fruit and vegetables, and macronutrients between the fitness groups. Only fibre was consumed significantly more by high-fit individuals (16.7 g *vs.* 12.1 g), but both groups were under the recommended daily intake. Generally, the habitual diet of the participants is reflective of the UK population's diet (HoM, 2009; NHS, 2020; Rauber et al., 2019), with a higher intake of sugar and fat and a low intake of fibre. For instance, all high-fit participants and 88.9% of low-fit participants exceeded the recommended daily sugar intake compared with 61.3% of the British population (Rauber et al., 2019). On the other hand, participants consumed less saturated fat (HF: 52.9%; LF: 33.3%) compared with >80% of the British population (HoM, 2009). Furthermore, 33% of the low-fit group consumed at least five portions of fruit and vegetables (average: 4 portions/day), which is in line with the

UK average of 28% (3.8 portions/day), while more individuals in the high-fit group (65%) reached the target recommendation (5.1 portions/day) (NHS, 2020). In regard to flavonoid consumption, participants in this study consumed 304–359 mg/day of flavonoids, which is less than what is estimated for the general population in the UK (655 mg/day) (Vogiatzoglou et al., 2015), but more than that of the US (234 mg/day) (Hu et al., 2024).

Based on the recent World Health Organization guidelines on physical activity (Bull et al., 2020), all individuals from both fitness groups met the minimum and conditional recommendations for the amount of time spent in moderate-intensity physical activity. Within the low-fit group, the recommended amount of time spent in vigorous intensity physical activity was only met by 26.7% and 6.7% of participants for the minimum and conditional recommendations, respectively. As might be expected, more individuals within the high-fit group met the minimum and conditional recommendations (76.5% and 58.8%, respectively) for the amount of time spent in vigorous intensity physical activity. Importantly, only the high-fit group appeared to have an adherence to physical activity similar to that of the population of England, in which 63.1% of adults aged 16 and over were considered physically active (GOV.UK, 2024). Finally, both fitness groups took more than 8,000 steps per day, which is above a threshold recently found to be associated with a progressively lower risk of mortality (Paluch et al., 2022). In summary, the habitual diet and levels of physical activity of our sample seem to generally reflect the population in the UK, so we consider these findings to have relevance for the general population.

### Limitations

One of the main limitations of the present study is that we included only young male adults, which means that the findings reported here are not applicable to females. A separate female-only dedicated study is needed to examine whether flavanol benefits and the impact of sitting is different depending on the phase of the menstrual cycle (early follicular *vs.* ovulatory phases): we hypothesise that flavanols may be more effective when oestrogen is available and, similarly, females might be more protected from the impact of sitting on endothelial function when oestrogen is high. Adding females to this cohort and focusing on a low-oestrogen phase of the menstrual cycle would regrettably miss a more nuanced/mechanistic understanding of how flavanol/sitting are modulated by cyclical hormonal changes. Future research should specifically examine the efficacy of flavanols during periods of sedentary behaviour in women only, considering the potential interplay with hormonal status and fitness. Secondly, we have split the sample into two different

groups (i.e. high fit, and low fit) only based on the maximal aerobic capacity from a $\dot{V}_{O_2\text{max}}$ test. Ideally, we could have also accounted for sport differences within our groups, considering that arterial remodelling (which can affect vascular function) is a process that is highly sport-specific (Rowley et al., 2012). Thirdly, we assessed participants' habitual diet using FFQs; however, we perceive that using food diaries instead would have been best in terms of characterising the participants' habitual diets. Finally, we opted to restrict the participants' diet for polyphenols 24 h prior to the testing session but it would have been ideal to restrict it for longer, considering that polyphenol metabolites may be detected in circulation for up to 80 h following ingestion (González-Sarrías et al., 2017).

## Conclusion

To the best of our knowledge, the present study is the first to investigate the acute effects of dietary flavanols on macro- and microvascular function during a prolonged period of sitting in young healthy men with distinct levels of fitness. We have shown for the first time that flavanols, when consumed just before a sitting period, are efficacious at preserving endothelial function in conduit arteries across upper- and lower-limb vasculature, without improving function at the downstream muscle microvasculature. Importantly, both high- and low-fit individuals benefited from intake of flavanols during sitting, highlighting that such a dietary strategy may be relevant for young males regardless of their levels of fitness. Additionally, both fitness groups experienced similar impairments in endothelial function, micro-vascular function and rises in diastolic BP during sitting, suggesting that better cardiorespiratory fitness may not be protective during periods of sitting. Given the high prevalence of sedentary time in young populations and its association with higher cardiovascular disease risk and hypertension, even when physical activity is taken into account, consuming high-flavanol foods (e.g. green or black tea, matcha, unprocessed cocoa, berries, apples) during sedentary periods may be used alone or in combination with other strategies (e.g. breaking up sitting) to reduce the impact of inactivity on the vascular system. Cocoa flavanols, in particular, have been shown to reduce the risk of cardiovascular events (Sesso et al., 2022), so using these strategically to target peri-ods during which the vasculature is more vulnerable (e.g. sitting) may enhance long-term health. Furthermore, exploring the combined efficacy of various flavonoids, including anthocyanin-rich berries, flavanol-rich green tea, and flavanone-rich citrus fruits, may be important to help translate this research into everyday life, given the evidence that these can be active in the vasculature within different time frames (Rendeiro et al., 2016; Rodriguez-Mateos et al., 2013). Specifically, the use of different flavonoid-rich foods may have the potential to confer protection during longer sitting bouts and allow flexibility for timing the intake in relation to sitting. Future work should focus on determining what is the maximum period during which flavonoids might be protective during sitting, what is the minimal effective dose and whether a combination of flavonoids from different sources could be more effective depending on the duration of the sitting period.

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

## Additional information

### Data availability statement

The data that support the findings of this study are available from the corresponding author (C.R.), upon reasonable request.

### Competing interests

None declared.

### Author contributions

C.R.: Conceptualisation and funding acquisition. A.D.: Data collection and data analysis. A.D.: Writing of original draft. C.R. and S.L.: Project administration, supervision, writing, review and editing.

## Funding

AD's PhD studentship was fully funded by the College of Life and Environmental Sciences, University of Birmingham (Birmingham, UK).

## Acknowledgements

The authors would like to thank Barry Callebaut (Leen Allegaert) for providing the high- and low-flavanol cocoa interventions. The graphical abstract was designed using resources from Flaticon.com.

## Keywords

cardiorespiratory fitness, cocoa flavanols, endothelial function, flow-mediated dilatation, sitting

## Supporting information

Additional supporting information can be found online in the Supporting Information section at the end of the HTML view of the article. Supporting information files available:

**Peer Review History**

