## [Peer Review History · The Journal of Physiology]

Dietary flavanols preserve upper- and lower-limb endothelial function during sitting in high- and low-fit young healthy males

Alessio Daniele, Samuel J.E. Lucas, and Catarina Rendeiro

DOI: 10.1113/JP289038

Corresponding author(s): Catarina Rendeiro (c.rendeiro@bham.ac.uk)

Review Timeline:

Submission Date:	10-Apr-2025
Editorial Decision:	07-Jul-2025
Revision Received:	29-Aug-2025
Accepted:	23-Sep-2025

Senior Editor: Harold Schultz

Reviewing Editor: Sophie Møller

Transaction Report:

Dear Dr Rendeiro,

Re: JP-RP-2025-289038 "**Dietary flavanols rescue upper- and lower-limb endothelial function during sitting in high- and low-fit young healthy males**" by Alessio Daniele, Samuel J.E. Lucas, and Catarina Rendeiro

Thank you for submitting your manuscript to The Journal of Physiology. It has been assessed by a Reviewing Editor and by 1 expert referee and we are pleased to tell you that it is potentially acceptable for publication following satisfactory major revision.

REVISION CHECKLIST:

Please upload two versions of your manuscript text: one with all relevant changes highlighted and one clean version with no

changes tracked. The manuscript file should include all tables and figure legends, but each figure/graph should be uploaded as separate, high-resolution files.

We look forward to receiving your revised submission.

Yours sincerely,

Harold Schultz
Senior Editor
The Journal of Physiology

REQUIRED ITEMS

- Author photo and profile. First or joint first authors are asked to provide a short biography (no more than 100 words for one author or 150 words in total for joint first authors) and a portrait photograph. These should be uploaded and clearly labelled together in a Word document with the revised version of the manuscript. See Information for Authors for further details.

- You must start the Methods section with a paragraph headed Ethical Approval. If experiments were conducted on humans, confirmation that informed consent was obtained, preferably in writing, that the studies conformed to the standards set by the latest revision of the Declaration of Helsinki and that the procedures were approved by a properly constituted ethics committee, which should be named, must be included in the article file. If the research study was registered (clause 35 of the Declaration of Helsinki), the registration database should be indicated, otherwise the lack of registration should be noted as an exception (e.g. The study conformed to the standards set by the Declaration of Helsinki, except for registration in a database). For further information see: <https://physoc.onlinelibrary.wiley.com/hub/human-experiments>.

- Please upload separate high-quality figure files via the submission form.

- Papers must comply with the Statistics Policy: https://jp.msubmit.net/cgi-bin/main.plex?form_type=display_requirements#statistics.

In summary:

- If $n \leq 30$, all data points must be plotted in the figure in a way that reveals their range and distribution. A bar graph with data points overlaid, a box and whisker plot or a violin plot (preferably with data points included) are acceptable formats.

- If $n > 30$, then the entire raw dataset must be made available either as supporting information, or hosted on a not-for-profit repository, e.g. FigShare, with access details provided in the manuscript.

- 'n' clearly defined (e.g. x cells from y slices in z animals) in the Methods. Authors should be mindful of pseudoreplication.

- All relevant 'n' values must be clearly stated in the main text, figures and tables.

- The most appropriate summary statistic (e.g. mean or median and standard deviation) must be used. Standard Error of the Mean (SEM) alone is not permitted.

- Exact p values must be stated. Authors must not use 'greater than' or 'less than'. Exact p values must be stated to three significant figures even when 'no statistical significance' is claimed.

- Please include an Abstract Figure file, as well as the Figure Legend text within the main article file. The Abstract Figure is a piece of artwork designed to give readers an immediate understanding of the research and should summarise the main conclusions. If possible, the image should be easily 'readable' from left to right or top to bottom. It should show the physiological relevance of the manuscript so readers can assess the importance and content of its findings. Abstract Figures should not merely recapitulate other figures in the manuscript. Please try to keep the diagram as simple as possible and without superfluous information that may distract from the main conclusion(s). Abstract Figures must be provided by authors no later than the revised manuscript stage and should be uploaded as a separate file during online submission labelled as File Type 'Abstract Figure'. Please also ensure that you include the figure legend in the main article file. All Abstract Figures should be created using BioRender. Authors should use The Journal's premium BioRender account to export high-resolution images. Details on how to use and access the premium account are included as part of this email.

EDITOR COMMENTS

Reviewing Editor:

Comments for Authors to ensure the paper complies with the Statistics Policy (Required):

Exact P-values are missing

Individual data points missing

Comments to the Author (Required):

Thank you for your submission to the Journal of Physiology. One expert reviewer has provided feedback to help you move further with your submission.

Please also see 'Required Items' above.

Aside from this, I have also reviewed your submission and have included my remarks below.

The submission by Daniele et al. investigates whether flavanol intake prior to a 2-hour sitting period can improve endothelial function in both the upper and lower limbs in individuals with varying fitness levels. This was explored in a randomised, counterbalanced, double-blind, cross-over, placebo-controlled study involving 40 young men aged 18-45 years. The authors conclude that flavanols consumed before a sitting period effectively restored endothelial function in conduit arteries of both the upper and lower limb vasculature.

A major point of concern in this study is the exclusive inclusion of male participants. I appreciate that you acknowledge the limitation of excluding women, and I understand the complexities associated with female physiology in clinical studies. However, I do not feel that the rationale provided sufficiently justifies this exclusion. While such an approach may have been historically common, the scientific community is increasingly moving away from the assumption that findings in men will automatically apply to women. Women constitute half of the global population, and methodological strategies exist to account for hormonal fluctuations. Including women, or at least making a more substantial effort toward inclusion, would have strengthened the study's relevance and impact. In fact, given the unique aspects of female physiology, a study in women could have been even more insightful than the one presented.

Beyond my major concern, I have some general comments for your consideration:

Statistical reporting: Please report exact p-values throughout the manuscript, including in the text, figures, and tables. Please include individual points in your graphs (as an overlay to the bar chart). Please refer to the Journal's author guidelines for the recommended statistical reporting format.

Terminology: For clarity and transparency, I recommend referring to your participants as "men" or "males" rather than "adults". The term "adult" can imply a mixed-sex population, which is not the case here. Additionally, a suggestion is to clarify whether flavanols prevent or rescue endothelial function. Since you only investigate flavanols prior to sitting your data suggests a preventative effect and not the rescue of endothelial function mentioned. "Rescue" is used in your title, key points, and in your abstract (here you use "ameliorate", which also does not imply prevention).

Translatability: Your study focuses on healthy young men. How do you anticipate these flavanol-mediated vascular effects might translate to older, more sedentary individuals, or to populations with chronic illness who experience prolonged sitting or bed rest? Would you expect an enhanced benefit, diminished response, or no effect beyond a certain age or after prolonged sedentary exposure? When is it too late?

Timing of flavanol administration: Could you clarify the rationale for administering flavanols prior to the sitting period? Would administration during or after the sitting period result in different vascular responses? Discussing why the pre-sitting timing was selected, and whether the intervention might also be protective or restorative if given at other time points, would strengthen the translational relevance of the findings.

I hope the feedback you have received will help you advance your manuscript further.

Senior Editor:

Comments for Authors to ensure the paper complies with the Statistics Policy (Required):

1. Provide the actual magnitude of the test statistic and the associated degrees of freedom or sample size in the main text, figures and/or tables. The exact p values must be stated to three significant figures even when 'no statistical significance' is being reported. As an exception, stating $p < 0.001$ is permitted. Do not use symbols or NS when possible.

2. Plot Individual data points in graphs whenever possible and in a way that reveals their range and distribution.

If the number of data points exceed the ability to be plotted legibly, use a violin plot and provide the entire raw dataset via a not-for-profit repository with access details.

3. State the statistical tests used in the figures and tables in their legends.

Senior Editor:

Thank you for submitting your research article for consideration to the Journal of Physiology. The article has been reviewed by experts in the field and found to require a revision to address all of the concerns raised. Please address all comments from the external referee and reviewing editor. Please also address the list of requirements for publication in the journal, including the statistical requirements.

REFeree COMMENTS

Referee #1:

This study sought to determine the impact of cocoa flavanols on vascular function in response to prolonged sitting in high and low fit young healthy individuals. Population: 40 young healthy males, stratified by VO₂ max. Low fit group was sig heavier and had higher diastolic BP and resting HR. Really strong characterization of participants PA levels, diets, etc. Main conclusions: 2 hours of sitting induced declines in FMD in both high and low fit individuals. The flavanol intervention prevented the sitting-induced decline in FMD in both groups. This is a well-done study with proper controls, strong study design, strong methodology, and thoughtful approach. The limitations are also nicely outlined in the discussion section. There is a concern regarding the novelty and significance of findings. A few additional minor concerns are outlined below.

Why is table 5 included if most of the data is presented again in graphs? Consider removing this table.

Presentation of data: Bar graphs should have individual dots for each data point for transparency and rigor. Also, consider presenting data as means \pm SEM instead of SD. It will improve the visibility of statistical differences.

Did the authors account for the baseline differences in body weight, BMI, diastolic BP, and HR for any of the statistical analyses?

END OF COMMENTS

REQUIRED ITEMS

- Author photo and profile. First or joint first authors are asked to provide a short biography (no more than 100 words for one author or 150 words in total for joint first authors) and a portrait photograph. These should be uploaded and clearly labelled together in a Word document with the revised version of the manuscript. See Information for Authors for further details.

First of all, thank you for all the comments provided. They are very insightful. We have tried to address all of them to the best of our ability. All the amendments in the text are reported in green.

Response: In accordance with your recommendations, a short biography and a portrait photograph of the first author have been uploaded with the revised version of the manuscript.

- You must start the Methods section with a paragraph headed Ethical Approval. If experiments were conducted on humans, confirmation that informed consent was obtained, preferably in writing, that the studies conformed to the standards set by the latest revision of the Declaration of Helsinki and that the procedures were approved by a properly constituted ethics committee, which should be named, must be included in the article file. If the research study was registered (clause 35 of the Declaration of Helsinki), the registration database should be indicated, otherwise the lack of registration should be noted as an exception (e.g. The study conformed to the standards set by the Declaration of Helsinki, except for registration in a database). For further information see: <https://physoc.onlinelibrary.wiley.com/hub/human-experiments>.

Response: We confirm that the requested information is already present within the Methods section, and can be found on line 135 of the manuscript.

From the manuscript:

MATERIALS AND METHODS

Ethical approval

The study was conducted in accordance with the Declaration of Helsinki, and was approved by the University of Birmingham Science, Technology, Engineering and Mathematics ethics committee (ERN_19-0851). Informed written consent was obtained from all participants before enrolment in the study.

- Please upload separate high-quality figure files via the submission form.

Response: As suggested, we have uploaded all figures separately via the submission form.

- Papers must comply with the Statistics Policy: https://jp.msubmit.net/cgi-bin/main.plex?form_type=display_requirements#statistics.

In summary:

- If n {less than or equal to} 30, all data points must be plotted in the figure in a way that reveals their range and distribution. A bar graph with data points overlaid, a box and whisker plot or a violin plot (preferably with data points included) are acceptable formats.
- If $n > 30$, then the entire raw dataset must be made available either as supporting information, or hosted on a not-for-profit repository, e.g. FigShare, with access details provided in the manuscript.
- ' n ' clearly defined (e.g. x cells from y slices in z animals) in the Methods. Authors should be mindful of pseudoreplication.
- All relevant ' n ' values must be clearly stated in the main text, figures and tables.
- The most appropriate summary statistic (e.g. mean or median and standard deviation) must be used. Standard Error of the Mean (SEM) alone is not permitted.
- Exact p values must be stated. Authors must not use 'greater than' or 'less than'. Exact p values must be stated to three significant figures even when 'no statistical significance' is claimed.

Response: Thank you for providing this information. In response to your suggestion, we have revised the bar and line graphs to incorporate individual data points, ensuring that the clarity and overall design of the figures remain uncompromised. Of note, semi-transparent rendering of individual data points was employed in the line graphs (Figure 7) to prevent overplotting and maintain the distinct visibility of the mean and standard deviation. This technique was chosen to ensure that the graphical representation effectively communicates both the distribution of individual observations and the central tendency, without compromising visual integrity.

For the allometrically scaled FMD figures (Figures 5C and 5D), individual data points are not displayed, given that the allometrically scaled FMD is estimated using a linear mixed model, in accordance with previous published methodologies (Atkinson & Batterham, 2013a, 2013b). This statistical approach, by its nature, generates summary statistics (mean and corrected standard deviation) which is what has been recommended for publications using this approach.

In addition, to enhance clarity and completeness, we have incorporated the sample size (N) for outcome measures such as SFA FMD and NIRS into all figures and tables captions.

We would like to also confirm that all our summary statistics use Mean and Standard Deviation.

Finally, we have now stated exact p values for all statistics provided, including non-significant values.

References

- Atkinson, G., & Batterham, A. M. (2013a). Allometric scaling of diameter change in the original flow-mediated dilation protocol. *Atherosclerosis*, 226(2), 425-427.

- Atkinson, G., & Batterham, A. M. (2013b). The percentage flow-mediated dilation index: a large-sample investigation of its appropriateness, potential for bias and causal nexus in vascular medicine. *Vascular Medicine*, 18(6), 354-365.

- Please include an Abstract Figure file, as well as the Figure Legend text within the main article file. The Abstract Figure is a piece of artwork designed to give readers an immediate understanding of the research and should summarise the main conclusions. If possible, the image should be easily 'readable' from left to right or top to bottom. It should show the physiological relevance of the manuscript so readers can assess the importance and content of its findings. Abstract Figures should not merely recapitulate other figures in the manuscript. Please try to keep the diagram as simple as possible and without superfluous information that may distract from the main conclusion(s). Abstract Figures must be provided by authors no later than the revised manuscript stage and should be uploaded as a separate file during online submission labelled as File Type 'Abstract Figure'. Please also ensure that you include the figure legend in the main article file. All Abstract Figures should be created using BioRender. Authors should use The Journal's premium BioRender account to export high-resolution images. Details on how to use and access the premium account are included as part of this email.

Response: We express our gratitude for access to The Journal's premium BioRender account, which facilitated the creation of the Abstract Figure. As per your suggestion, the Abstract Figure has been submitted as a separate file alongside the updated manuscript. The legend of the Abstract Figure is provided at the end of the manuscript in the List of Figures, as recommended.

EDITOR COMMENTS

Reviewing Editor:

Comments for Authors to ensure the paper complies with the Statistics Policy (Required):

Exact P-values are missing

Individual data points missing

Response: Thanks for your comment. We can confirm that we now present exact *p*-values, *F*-values and degrees of freedom in our statistics. As specified above, our Figures contain now individual data points. All our summary statistics use Mean and Standard Deviation, so we have ensured to the best of our ability full compliance with the established Statistics Policy of the journal.

Comments to the Author (Required):

Thank you for your submission to the Journal of Physiology. One expert reviewer has provided feedback to help you move further with your submission.

Please also see 'Required Items' above.

Aside from this, I have also reviewed your submission and have included my remarks below.

The submission by Daniele et al. investigates whether flavanol intake prior to a 2-hour sitting period can improve endothelial function in both the upper and lower limbs in individuals with varying fitness levels. This was explored in a randomised, counterbalanced, double-blind, cross-over, placebo-controlled study involving 40 young men aged 18-45 years. The authors conclude that flavanols consumed before a sitting period effectively restored endothelial function in conduit arteries of both the upper and lower limb vasculature.

A major point of concern in this study is the exclusive inclusion of male participants. I appreciate that you acknowledge the limitation of excluding women, and I understand the complexities associated with female physiology in clinical studies. However, I do not feel that the rationale provided sufficiently justifies this exclusion. While such an approach may have been historically common, the scientific community is increasingly moving away from the assumption that findings in men will automatically apply to women. Women constitute half of the global population, and methodological strategies exist to account for hormonal fluctuations. Including women, or at least making a more substantial effort toward inclusion, would have strengthened the study's relevance and impact. In fact, given the unique aspects of female physiology, a study in women could have been even more insightful than the one presented.

Response: Thank you very much for your insightful comment. We could not agree more with your comment: having a study looking at women exclusively was (ironically) part of our initial plan for this PhD project. At the time, we felt like having 2 separate studies, one focused on males and one focused on females would make most sense. Our rationale was that a separate study in females would allow us to investigate these effects at 2 distinct points in the menstrual cycle, in early follicular phase and in the ovulatory phase. This would be important given that we hypothesised that flavanols might be more effective in the ovulation phase, when oestrogen is more available (from previous preliminary unpublished data). On the other hand, we also hypothesized that women during their ovulation phase would be more protected from sitting and experience minimal declines in vascular function compared to the early follicular phase. These were questions that could only be addressed in a women-only study, hence our choice to have this current study focus on males. Unfortunately, the males' study was severely affected by the COVID-19 lockdowns, we lost a considerable amount of data and time. In reality, this meant that delivering a female study would be difficult given the time constraints. We made decision to redirect our efforts to a more feasible study that could be completed within the PhD time. We completed a similar

study in 20 older adults (male and female participants), given that this population that spends a considerable amount of time in sitting activities and most likely at higher risk of disease from sedentary behaviour. This work is currently under consideration at another journal.

We fully acknowledge that the exclusive focus on male participants in this study represents a limitation, but it was never assumed that our findings in males would automatically apply to females. We had a scientific rationale and specific hypothesis that could only be tested in two separate cohorts, and we believed would allow us for a more comprehensive understanding of how flavanol/sitting may be influenced by hormonal changes. In retrospective we could have just included females as part of this cohort and have a more comprehensive study. However, we are currently pursuing funding to continue this work in females.

To the reviewer's point, we absolutely agree that the current findings are not directly generalizable to the female population, so we have expanded our limitation section to ensure we give this point full attention.

From our limitation section:

One of the main limitations of the present study is that we included only young male adults, which means that the findings reported here are not applicable to females. A separate female-only dedicated study is needed to examine whether flavanol benefits and the impact of sitting is different depending on the phase of the menstrual cycle (early follicular vs. ovulatory phases): we hypothesize that flavanols may be more effective when oestrogen is available and similarly, females might be more protected from the impact of sitting on endothelial function when oestrogen is high. Adding females to this cohort and focusing on a low-oestrogen phase of the menstrual cycle would regrettably miss a more nuanced/mechanistic understanding of how flavanol/sitting are modulated by cyclical hormonal changes. Future research should specifically examine the efficacy of flavanols during periods of sedentary behaviour in women only, considering the potential interplay with hormonal status and fitness.

Additionally, we have also modified our methods/exclusion criteria to more clearly reflect the reasoning underlying a male-only study.

From the participant's section:

Females were not included, as a separate female-only RCT is required to ensure that interactions with distinct stages of menstrual cycle are addressed, given the rationale for differential modulation of vascular function by flavanols depending on the levels of circulatory oestrogen (Moreno-Ulloa et al., 2015; Moreno-Ulloa et al., 2018).

We truly believe that even though this data is in males only, it is the first to investigate the impact of dietary approach in the context of sitting, it is also the first that compares high and low fit individuals, which also gave us important information about how sitting alone may affect individuals with distinct cardiorespiratory fitness. We believe this data will be valuable to inform future studies in this area. We hope this is acceptable.

Beyond my major concern, I have some general comments for your consideration:

Statistical reporting: Please report exact p-values throughout the manuscript, including in the text, figures, and tables. Please include individual points in your graphs (as an overlay to the bar chart). Please refer to the Journal's author guidelines for the recommended statistical reporting format.

Response: As previously indicated, we can confirm that we now present exact *p*-values, *F*-values and degrees of freedom in our statistics. As specified above, our Figures contain now individual data points. All our summary statistics use Mean and Standard Deviation, so we have ensured to the best of our ability full compliance with the established Statistics Policy of the journal.

Terminology: For clarity and transparency, I recommend referring to your participants as "men" or "males" rather than "adults". The term "adult" can imply a mixed-sex population, which is not the case here. Additionally, a suggestion is to clarify whether flavanols prevent or rescue endothelial function. Since you only investigate flavanols prior to sitting your data suggests a preventative effect and not the rescue of endothelial function mentioned. "Rescue" is used in your title, key points, and in your abstract (here you use "ameliorate", which also does not imply prevention).

Response: Thank you for your comment. We acknowledge your point regarding the terminology. We have thoroughly re-evaluated the terms in question and implemented revisions where deemed necessary to enhance clarity and precision, thereby mitigating potential ambiguity.

Translatability: Your study focuses on healthy young men. How do you anticipate these flavanol-mediated vascular effects might translate to older, more sedentary individuals, or to populations with chronic illness who experience prolonged sitting or bed rest? Would you expect an enhanced benefit, diminished response, or no effect beyond a certain age or after prolonged sedentary exposure? When is it too late?

Response: Regarding your queries, we conducted a separate study on a different sample of healthy older adults, over 65 yr. of age, (included males and females), and observed beneficial effects of flavanol intake prior to sitting. These were very consistent and of similar magnitude to the benefits observed in the present study, even though baseline FMD were lower for the older group, as expected. This indicates that indeed foods rich in flavanols might be effective in protecting the vascular system in old age. This was however a relatively healthy older cohort, future studies need to address this question in populations at higher risk of disease.

Additionally, there is very well-established research indicating positive effects on endothelial function after flavanol intake (acute and chronic) in at-risk populations and populations with disease (e.g., hypertension, PAD, metabolic syndrome, obesity) (Balzer et al., 2008; Bapir et al., 2022; Davison et al., 2008; Grassi et al., 2005; Heiss et al., 2010; Njike et al., 2011). It is plausible to hypothesise that flavanols could confer similar benefits to these populations, potentially mitigating some of the adverse vascular impact associated with prolonged sedentary behaviour.

Ascertaining whether flavanol effectiveness declines in advanced age (e.g., individuals over 75 years) is challenging, as direct experimental studies addressing this specific demographic in the context of sitting are absent. Our study in older adults included individuals up to 87 years old, and they seem to still benefit. In support of this, there is robust evidence suggesting the efficacy of flavanols in individuals older than 60 years old (Gröne et al., 2019; Heiss et al., 2015; Monahan et al., 2011), but no studies looked at individuals over 80 years old, for example. Generally, it seems that these compounds can be effective even when the vasculature is more deteriorated, such as in old age and diseased states.

References

- Balzer, J., Rassaf, T., Heiss, C., Kleinbongard, P., Lauer, T., Merx, M., ... & Kelm, M. (2008). Sustained benefits in vascular function through flavanol-containing cocoa in medicated diabetic patients: a double-masked, randomized, controlled trial. *Journal of the American college of cardiology*, 51(22), 2141-2149.
- Bapir, M., Untracht, G. R., Cooke, D., McVey, J. H., Skene, S. S., Campagnolo, P., ... & Heiss, C. (2022). Cocoa flavanol consumption improves lower extremity endothelial function in healthy individuals and people with type 2 diabetes. *Food & function*, 13(20), 10439-10448.
- Davison, K., Coates, A. M., Buckley, J. D., & Howe, P. R. C. (2008). Effect of cocoa flavanols and exercise on cardiometabolic risk factors in overweight and obese subjects. *International journal of obesity*, 32(8), 1289-1296.
- Grassi, D., Necozione, S., Lippi, C., Croce, G., Valeri, L., Pasqualetti, P., ... & Ferri, C. (2005). Cocoa reduces blood pressure and insulin resistance and improves endothelium-dependent vasodilation in hypertensives. *Hypertension*, 46(2), 398-405.
- Gröne, M., Sansone, R., Höffken, P., Horn, P., Rodriguez-Mateos, A., Schroeter, H., ... & Heiss, C. (2019). Cocoa flavanols improve endothelial functional integrity in healthy young and elderly subjects. *Journal of agricultural and food chemistry*, 68(7), 1871-1876.
- Heiss, C., Jahn, S., Taylor, M., Real, W. M., Angeli, F. S., Wong, M. L., ... & Yeghiazarians, Y. (2010). Improvement of endothelial function with dietary flavanols is associated with mobilization of circulating angiogenic cells in patients with coronary artery disease. *Journal of the American College of Cardiology*, 56(3), 218-224.
- Heiss, C., Sansone, R., Karimi, H., Krabbe, M., Schuler, D., Rodriguez-Mateos, A., ... & FLAVIOLA Consortium, European Union 7th Framework Program. (2015). Impact of cocoa flavanol intake on age-dependent vascular stiffness in healthy men: a randomized, controlled, double-masked trial. *Age*, 37(3), 56.
- Monahan, K. D., Feehan, R. P., Kunselman, A. R., Preston, A. G., Miller, D. L., & Lott, M. E. (2011). Dose-dependent increases in flow-mediated dilation following acute cocoa ingestion in healthy older adults. *Journal of applied physiology*, 111(6), 1568-1574.
- Njike, V. Y., Faridi, Z., Shuval, K., Dutta, S., Kay, C. D., West, S. G., ... & Katz, D. L. (2011). Effects of sugar-sweetened and sugar-free cocoa on endothelial function in overweight adults. *International journal of cardiology*, 149(1), 83-88.

Timing of flavanol administration: Could you clarify the rationale for administering flavanols prior to the sitting period? Would administration during or after the sitting period result in

different vascular responses? Discussing why the pre-sitting timing was selected, and whether the intervention might also be protective or restorative if given at other time points, would strengthen the translational relevance of the findings.

Response: Flavanol metabolites are detectable in circulation within 1 hour of intake, typically peaking around 2 hours post-ingestion followed by a decline and clearance by 4 hours. A subsequent wave of gut-derived flavanol metabolites can then be observed from around 6–7 h, persisting in circulation for up to 48 hours. Our rationale for administering cocoa flavanols before the 2-hour sitting intervention was to ensure sustained flavanol availability throughout the sitting period and testing whether the presence of these compounds in circulation would be helpful during sitting.

We propose that flavanol administration during, rather than strictly prior to, sedentary periods may still confer benefits, particularly during extended sitting durations. For example, benefits in endothelial function after flavanol intake are detected within 1 h, which means that taking flavanols during a sitting bout might still be effective. Furthermore, other more rapidly acting flavonoid-rich foods, such as berries (e.g., blueberry anthocyanin metabolites peak in 30 min), could offer advantages both during and immediately following sedentary activity to help the recovery of endothelial function post sitting.

Importantly, movement-based interventions (e.g. standing, fidgeting, walking) are demonstrably effective in reversing impairments in vascular function, including declines in FMD, after prolonged sitting. Moving/Standing alone appears to be a sufficient stimulus for recovery of endothelial function, so dietary interventions such as the one tested in this study may be more helpful before or during the sitting bouts to prevent declines in function during sitting. Exercise and movement should be the go-to strategy, but when those approaches are not possible, using additional dietary strategies may be a good strategy.

We have now added to the methods, a justification for a pre-sitting administration of the flavanol intervention:

Flavanols were administered immediately prior to sitting to ensure their bioavailability during the 2-hour sitting period, as flavanols are known to be bioavailable within 1 hour of intake, peaking at 2 hours post-ingestion (Schroeter et al., 2006).

We have also revised the conclusion section to strengthen the translational relevance of the findings.

Conclusion:

Furthermore, exploring the combined efficacy of various flavonoids, including anthocyanin-rich berries, flavanol-rich green tea, and flavanone-rich citrus fruits, may be important to help translate this research into everyday life, given the evidence that these can be active in the vasculature within different time frames (Rendeiro et al., 2016; Rodriguez-Mateos et al., 2013). Specifically, the use of different flavonoid-rich foods may have the potential to confer protection during longer sitting bouts and allow flexibility for timing of intake in relation to sitting. Future work should focus on determining what is the maximum period during which flavonoids might be protective during sitting, what is the minimal effective dose and whether a combination of flavonoids from different sources can be more effective depending on the duration of the sitting period.

I hope the feedback you have received will help you advance your manuscript further.

Senior Editor:

Comments for Authors to ensure the paper complies with the Statistics Policy (Required):

1. Provide the actual magnitude of the test statistic and the associated degrees of freedom or sample size in the main text, figures and/or tables. The exact p values must be stated to three significant figures even when 'no statistical significance' is being reported. As an exception, stating $p < 0.001$ is permitted. Do not use symbols or NS when possible.

Response: As previously indicated, the authors have ensured full compliance with the established Statistics Policy, specifically regarding the presentation of exact p -values, F -statistics, individual data points, and the associated degrees of freedom.

2. Plot Individual data points in graphs whenever possible and in a way that reveals their range and distribution.

If the number of data points exceed the ability to be plotted legibly, use a violin plot and provide the entire raw dataset via a not-for-profit repository with access details.

Response: As previously indicated, we included individual data points in the bar and line graphs, with the exception of the 'allometrically scaled FMD' graphs (discussed above in a previous question).

3. State the statistical tests used in the figures and tables in their legends.

Response: As suggested, the relevant information has been integrated into the legends of the figures and tables.

As reported:

Three-way repeated measures ANOVA conducted. Bonferroni post hoc for significant interactions.

Senior Editor:

Thank you for submitting your research article for consideration to the Journal of Physiology. The article has been reviewed by experts in the field and found to require a revision to address all of the concerns raised. Please address all comments from the external referee and reviewing editor. Please also address the list of requirements for publication in the journal, including the statistical requirements.

REFEREE COMMENTS

Referee #1:

This study sought to determine the impact of cocoa flavanols on vascular function in response to prolonged sitting in high and low fit young healthy individuals. Population: 40 young healthy males, stratified by VO₂ max. Low fit group was sig heavier and had higher diastolic BP and resting HR. Really strong characterization of participants PA levels, diets, etc. Main conclusions: 2 hours of sitting induced declines in FMD in both high and low fit individuals. The flavanol intervention prevented the sitting-induced decline in FMD in both groups. This is a well-done study with proper controls, strong study design, strong methodology, and thoughtful approach. The limitations are also nicely outlined in the discussion section. There is a concern regarding the novelty and significance of findings. A few additional minor concerns are outlined below.

Why is table 5 included if most of the data is presented again in graphs? Consider removing this table.

Response: We appreciate your suggestion and agree that the most interesting and key outcomes are presented in graphical form. Table 5 has now been removed from the manuscript as suggested. We still report effects on parameters such as retrograde shear rate and blood flow, as well as arterial diameter in the text, indicating 'data not shown'.

Presentation of data: Bar graphs should have individual dots for each data point for transparency and rigor. Also, consider presenting data as means +/- SEM instead of SD. It will improve the visibility of statistical differences.

Response: As previously indicated, we included individual data points in the bar and line graphs, with the exception of the 'allometrically scaled FMD' Figures (discussed above in a previous question).

In regard to your second point, we wish to reiterate that the application of standard deviation (SD) in our analysis adheres to the statistical guidelines outlined by the journal, as stated:

Data summaries should be presented as mean (SD) with a clear statement of n; and presented in the main text, figures and their legends and tables. Standard Deviation (SD) must be used instead of Standard Error of the Mean (SEM), unless the use of SEM is fully justified and reported alongside confidence intervals.

Did the authors account for the baseline differences in body weight, BMI, diastolic BP, and HR for any of the statistical analyses?

Response: Thank you for your constructive comment. We have followed your suggestion and have conducted the statistical analysis for the FMD outcomes incorporating each one of these parameters as covariates. We observed they consistently come out as non-significant, and for that reason we decided to remove the covariates from our analysis. Importantly, we noticed that when we include these parameters as covariates, it eliminates some of the differences in fitness that we had previously detected. This indicates that, by including these covariates, we are removing some of the variability that distinguishes these groups, which is core to our research question. We hope this is acceptable.

Dear Dr Rendeiro,

Re: JP-RP-2025-289038R1 **"Dietary flavanols preserve upper- and lower-limb endothelial function during sitting in high- and low-fit young healthy males"** by Alessio Daniele, Samuel J.E. Lucas, and Catarina Rendeiro

We are pleased to tell you that your paper has been accepted for publication in The Journal of Physiology.

Yours sincerely,

Harold Schultz
Senior Editor
The Journal of Physiology

If you would like to receive our 'Research Roundup', a monthly newsletter highlighting the cutting-edge research published in The Physiological Society's family of journals (The Journal of Physiology, Experimental Physiology, Physiological Reports, The Journal of Nutritional Physiology and The Journal of Precision Medicine: Health and Disease), please click this link, fill in your name and email address and select 'Research Roundup':
<https://www.physoc.org/journals-and-media/membernews>

- **TRANSPARENT PEER REVIEW POLICY:** To improve the transparency of its peer review process, The Journal of Physiology publishes online as supporting information the peer review history of all articles accepted for publication. Readers will have access to decision letters, including Editors' comments and referee reports, for each version of the manuscript as well as any author responses to peer review comments. Referees can decide whether or not they wish to be named on the peer review history document.
- You can help your research get the attention it deserves! Check out Wiley's free Promotion Guide for best-practice recommendations for promoting your work at: www.wileyauthors.com/eeo/guide. You can learn more about Wiley Editing Services which offers professional video, design, and writing services to create shareable video abstracts, infographics, conference posters, lay summaries, and research news stories for your research at: www.wileyauthors.com/eeo/promotion.
- **IMPORTANT NOTICE ABOUT OPEN ACCESS:** To assist authors whose funding agencies mandate public access to published research findings sooner than 12 months after publication, The Journal of Physiology allows authors to pay an Open Access (OA) fee to have their papers made freely available immediately on publication.

EDITOR COMMENTS

Reviewing Editor:

Thank you for the revisions made and for addressing all of the comments that have been provided to you. I have no further comments to the revision.

Senior Editor:

The editors thank the authors for the final adjustments to the manuscript. The article is now accepted for publication. Congratulations on an interesting and insightful study. Please consider the Journal of Physiology for your future studies.

REFEREE COMMENTS

Referee #1:

The authors have been very responsive to previous comments. I have no further comments.